# CERTIFYING SOME DISTRIBUTIONAL ROBUSTNESS WITH PRINCIPLED ADVERSARIAL TRAINING

**Aman Sinha**[*,1]**, Hongseok Namkoong**[*,2]**, John Duchi**[1,3]
Departments of [1]Electrical Engineering, [2]Management Science & Engineering, [3]Statistics
Stanford University
Stanford, CA 94305
{amans,hnamk,jduchi}@stanford.edu

## ABSTRACT

Neural networks are vulnerable to adversarial examples and researchers have proposed many heuristic attack and defense mechanisms. We address this problem through the principled lens of distributionally robust optimization, which guarantees performance under adversarial input perturbations. By considering a Lagrangian penalty formulation of perturbing the underlying data distribution in a Wasserstein ball, we provide a training procedure that augments model parameter updates with worst-case perturbations of training data. For smooth losses, our procedure provably achieves moderate levels of robustness with little computational or statistical cost relative to empirical risk minimization. Furthermore, our statistical guarantees allow us to efficiently certify robustness for the population loss. For imperceptible perturbations, our method matches or outperforms heuristic approaches.

## 1 INTRODUCTION

Consider the classical supervised learning problem, in which we minimize an expected loss $\mathbb{E}_{P_0}[\ell(\theta; Z)]$ over a parameter $\theta \in \Theta$, where $Z \sim P_0$, $P_0$ is a distribution on a space $\mathcal{Z}$, and $\ell$ is a loss function. In many systems, robustness to changes in the data-generating distribution $P_0$ is desirable, whether they be from covariate shifts, changes in the underlying domain (Ben-David et al., 2010), or adversarial attacks (Goodfellow et al., 2015; Kurakin et al., 2016). As deep networks become prevalent in modern performance-critical systems (perception for self-driving cars, automated detection of tumors), model failure is increasingly costly; in these situations, it is irresponsible to deploy models whose robustness and failure modes we do not understand or cannot certify.

Recent work shows that neural networks are vulnerable to adversarial examples; seemingly imperceptible perturbations to data can lead to misbehavior of the model, such as misclassification of the output (Goodfellow et al., 2015; Nguyen et al., 2015; Kurakin et al., 2016; Moosavi-Dezfooli et al., 2016). Consequently, researchers have proposed adversarial attack and defense mechanisms (Papernot et al., 2016a;b;c; Rozsa et al., 2016; Carlini & Wagner, 2017; He et al., 2017; Madry et al., 2017; Tramèr et al., 2017). These works provide an initial foundation for adversarial training, but it is challenging to rigorously identify the classes of attacks against which they can defend (or if they exist). Alternative approaches that provide formal verification of deep networks (Huang et al., 2017; Katz et al., 2017a;b) are NP-hard in general; they require prohibitive computational expense even on small networks. Recently, researchers have proposed convex relaxations of the NP-hard verification problem with some success (Kolter & Wong, 2017; Raghunathan et al., 2018), though they may be difficult to scale to large networks. In this context, our work is situated between these agendas: we develop efficient procedures with rigorous guarantees for *small to moderate* amounts of robustness.

We take the perspective of distributionally robust optimization and provide an adversarial training procedure with provable guarantees on its computational and statistical performance. We postulate a class $\mathcal{P}$ of distributions around the data-generating distribution $P_0$ and consider the problem

$$\underset{\theta \in \Theta}{\text{minimize}} \sup_{P \in \mathcal{P}} \mathbb{E}_P[\ell(\theta; Z)]. \tag{1}$$

---

[*]Equal contribution

The choice of $\mathcal{P}$ influences robustness guarantees and computability; we develop robustness sets $\mathcal{P}$ with computationally efficient relaxations that apply even when the loss $\ell$ is non-convex. We provide an adversarial training procedure that, for smooth $\ell$, enjoys convergence guarantees similar to non-robust approaches while *certifying* performance even for the worst-case population loss $\sup_{P \in \mathcal{P}} \mathbb{E}_P[\ell(\theta; Z)]$. On a simple implementation in Tensorflow, our method takes 5–10$\times$ as long as stochastic gradient methods for empirical risk minimization (ERM), matching runtimes for other adversarial training procedures (Goodfellow et al., 2015; Kurakin et al., 2016; Madry et al., 2017). We show that our procedure—which learns to protect against adversarial perturbations in the training dataset—generalizes, allowing us to train a model that prevents attacks to the test dataset.

We briefly overview our approach. Let $c : \mathcal{Z} \times \mathcal{Z} \to \mathbb{R}_+ \cup \{\infty\}$, where $c(z, z_0)$ is the "cost" for an adversary to perturb $z_0$ to $z$ (we typically use $c(z, z_0) = \|z - z_0\|_p^2$ with $p \geq 1$). We consider the robustness region $\mathcal{P} = \{P : W_c(P, P_0) \leq \rho\}$, a $\rho$-neighborhood of the distribution $P_0$ under the Wasserstein metric $W_c(\cdot, \cdot)$ (see Section 2 for a formal definition). For deep networks and other complex models, this formulation of problem (1) is intractable with arbitrary $\rho$. Instead, we consider its Lagrangian relaxation for a fixed penalty parameter $\gamma \geq 0$, resulting in the reformulation

$$\operatorname*{minimize}_{\theta \in \Theta} \left\{ F(\theta) := \sup_P \left\{ \mathbb{E}_P[\ell(\theta; Z)] - \gamma W_c(P, P_0) \right\} = \mathbb{E}_{P_0}[\phi_\gamma(\theta; Z)] \right\} \qquad (2a)$$

$$\text{where} \quad \phi_\gamma(\theta; z_0) := \sup_{z \in \mathcal{Z}} \left\{ \ell(\theta; z) - \gamma c(z, z_0) \right\}. \qquad (2b)$$

(See Proposition 1 for a rigorous statement of these equalities.) Here, we have replaced the usual loss $\ell(\theta; Z)$ by the robust surrogate $\phi_\gamma(\theta; Z)$; this surrogate (2b) allows adversarial perturbations of the data $z$, modulated by the penalty $\gamma$. We typically solve the penalty problem (2) with $P_0$ replaced by the empirical distribution $\widehat{P}_n$, as $P_0$ is unknown (we refer to this as the penalty problem below).

The key feature of the penalty problem (2) is that moderate levels of robustness—in particular, defense against imperceptible adversarial perturbations—are achievable at essentially no computational or statistical cost for *smooth losses* $\ell$. Specifically, for large enough penalty $\gamma$ (by duality, small enough robustness $\rho$), the function $z \mapsto \ell(\theta; z) - \gamma c(z, z_0)$ in the robust surrogate (2b) is strongly concave and hence *easy* to optimize if $\ell(\theta, z)$ is smooth in $z$. Consequently, stochastic gradient methods applied to problem (2) have similar convergence guarantees as for non-robust methods (ERM). In Section 3, we provide a *certificate of robustness* for any $\rho$; we give an efficiently computable data-dependent upper bound on the worst-case loss $\sup_{P : W_c(P, P_0) \leq \rho} \mathbb{E}_P[\ell(\theta; Z)]$. That is, the worst-case performance of the output of our principled adversarial training procedure is guaranteed to be no worse than this certificate. Our bound is tight when $\rho = \widehat{\rho}_n$, the achieved robustness for the empirical objective. These results suggest advantages of networks with smooth activations rather than ReLU's. We experimentally verify our results in Section 4 and show that we match or achieve state-of-the-art performance on a variety of adversarial attacks.

**Robust optimization and adversarial training**    The standard robust-optimization approach minimizes losses of the form $\sup_{u \in \mathcal{U}} \ell(\theta; z + u)$ for some uncertainty set $\mathcal{U}$ (Ratliff et al., 2006; Ben-Tal et al., 2009; Xu et al., 2009). Unfortunately, this approach is intractable except for specially structured losses, such as the composition of a linear and simple convex function (Ben-Tal et al., 2009; Xu et al., 2009; 2012). Nevertheless, this robust approach underlies recent advances in adversarial training (Szegedy et al., 2013; Goodfellow et al., 2015; Papernot et al., 2016b; Carlini & Wagner, 2017; Madry et al., 2017), which heuristically perturb data during a stochastic optimization procedure.

One such heuristic uses a locally linearized loss function (proposed with $p = \infty$ as the "fast gradient sign method" (Goodfellow et al., 2015)):

$$\Delta_{x_i}(\theta) := \operatorname*{argmax}_{\|\eta\|_p \leq \epsilon} \{\nabla_x \ell(\theta; (x_i, y_i))^T \eta\} \quad \text{and perturb} \quad x_i \to x_i + \Delta_{x_i}(\theta). \qquad (3)$$

One form of adversarial training trains on the losses $\ell(\theta; (x_i + \Delta_{x_i}(\theta), y_i))$ (Goodfellow et al., 2015; Kurakin et al., 2016), while others perform iterated variants (Papernot et al., 2016b; Carlini & Wagner, 2017; Madry et al., 2017; Tramèr et al., 2017). Madry et al. (2017) observe that these procedures attempt to optimize the objective $\mathbb{E}_{P_0}[\sup_{\|u\|_p \leq \epsilon} \ell(\theta; Z + u)]$, a constrained version of the penalty problem (2). This notion of robustness is typically intractable: the inner supremum is generally non-concave in $u$, so it is unclear whether model-fitting with these techniques converges,

and there are possibly worst-case perturbations these techniques do not find. Indeed, it is NP-hard to find worst-case perturbations when deep networks use ReLU activations, suggesting difficulties for fast and iterated heuristics (see Lemma 2 in Appendix B). Smoothness, which can be obtained in standard deep architectures with exponential linear units (ELU's) (Clevert et al., 2015), allows us to find Lagrangian worst-case perturbations with low computational cost.

**Distributionally robust optimization** To situate the current work, we review some of the substantial body of work on robustness and learning. The choice of $\mathcal{P}$ in the robust objective (1) affects both the richness of the uncertainty set we wish to consider as well as the tractability of the resulting optimization problem. Previous approaches to distributional robustness have considered finite-dimensional parametrizations for $\mathcal{P}$, such as constraint sets for moments, support, or directional deviations (Chen et al., 2007; Delage & Ye, 2010; Goh & Sim, 2010), as well as non-parametric distances for probability measures such as $f$-divergences (Ben-Tal et al., 2013; Bertsimas et al., 2013; Lam & Zhou, 2015; Miyato et al., 2015; Duchi et al., 2016; Namkoong & Duchi, 2016), and Wasserstein distances (Esfahani & Kuhn, 2015; Shafieezadeh-Abadeh et al., 2015; Blanchet et al., 2016). In contrast to $f$-divergences (e.g. $\chi^2$- or Kullback-Leibler divergences) which are effective when the support of the distribution $P_0$ is fixed, a Wasserstein ball around $P_0$ includes distributions $Q$ with different support and allows (in a sense) robustness to unseen data.

Many authors have studied tractable classes of uncertainty sets $\mathcal{P}$ and losses $\ell$. For example, Ben-Tal et al. (2013) and Namkoong & Duchi (2017) use convex optimization approaches for $f$-divergence balls. For worst-case regions $\mathcal{P}$ formed by Wasserstein balls, Esfahani & Kuhn (2015), Shafieezadeh-Abadeh et al. (2015), and Blanchet et al. (2016) show how to convert the saddle-point problem (1) to a regularized ERM problem, but this is possible only for a limited class of convex losses $\ell$ and costs $c$. In this work, we treat a much larger class of losses and costs and provide direct solution methods for a Lagrangian relaxation of the saddle-point problem (1). One natural application area is in domain adaptation (Lee & Raginsky, 2017); concurrently with this work, Lee & Raginsky provide guarantees similar to ours for the empirical minimizer of the robust saddle-point problem (1) and give specialized bounds for domain adaptation problems. In contrast, our approach is to use the distributionally robust approach to both defend against imperceptible adversarial perturbations and develop efficient optimization procedures.

## 2 PROPOSED APPROACH

Our approach is based on the following simple insight: assume that the function $z \mapsto \ell(\theta; z)$ is smooth, meaning there is some $L$ for which $\nabla_z \ell(\theta; \cdot)$ is $L$-Lipschitz. Then for any $c : \mathcal{Z} \times \mathcal{Z} \to \mathbb{R}_+ \cup \{\infty\}$ 1-strongly convex in its first argument, a Taylor expansion yields

$$\ell(\theta; z') - \gamma c(z', z_0) \leq \ell(\theta; z) - \gamma c(z, z_0) + \langle \nabla_z(\ell(\theta; z) - \gamma c(z, z_0)), z' - z \rangle + \frac{L - \gamma}{2} \|z - z'\|_2^2.$$
(4)

For $\gamma \geq L$ this is the first-order condition for $(\gamma - L)$-strong concavity of $z \mapsto (\ell(\theta; z) - \gamma c(z, z_0))$. Thus, whenever the loss is smooth enough in $z$ and the penalty $\gamma$ is large enough (corresponding to less robustness), computing the surrogate (2b) is a strongly-concave optimization problem.

We leverage the insight (4) to show that as long as we do not require *too much* robustness, this strong concavity approach (4) provides a computationally efficient and principled approach for robust optimization problems (1). Our starting point is a duality result for the minimax problem (1) and its Lagrangian relaxation for Wasserstein-based uncertainty sets, which makes the connections between distributional robustness and the "lazy" surrogate (2b) clear. We then show (Section 2.1) how stochastic gradient descent methods can efficiently find minimizers (in the convex case) or approximate stationary points (when $\ell$ is non-convex) for our relaxed robust problems.

**Wasserstein robustness and duality** Wasserstein distances define a notion of closeness between distributions. Let $\mathcal{Z} \subset \mathbb{R}^m$, and let $(\mathcal{Z}, \mathcal{A}, P_0)$ be a probability space. Let the transportation cost $c : \mathcal{Z} \times \mathcal{Z} \to [0, \infty)$ be nonnegative, lower semi-continuous, and satisfy $c(z, z) = 0$. For example, for a differentiable convex $h : \mathcal{Z} \to \mathbb{R}$, the Bregman divergence $c(z, z_0) = h(z) - h(z_0) - \langle \nabla h(z_0), z - z_0 \rangle$ satisfies these conditions. For probability measures $P$ and $Q$ supported on $\mathcal{Z}$, let $\Pi(P, Q)$ denote their couplings, meaning measures $M$ on $\mathcal{Z}^2$ with $M(A, \mathcal{Z}) = P(A)$ and

$M(\mathcal{Z}, A) = Q(A)$. The Wasserstein distance between $P$ and $Q$ is

$$W_c(P, Q) := \inf_{M \in \Pi(P,Q)} \mathbb{E}_M[c(Z, Z')].$$

For $\rho \geq 0$ and distribution $P_0$, we let $\mathcal{P} = \{P : W_c(P, P_0) \leq \rho\}$, considering the Wasserstein form of the robust problem (1) and its Lagrangian relaxation (2) with $\gamma \geq 0$. The following duality result (Blanchet & Murthy, 2016) gives the equality (2) for the relaxation and an analogous result for the problem (1). We give an alternative proof in Appendix C.1 for convex, continuous cost functions.

**Proposition 1.** *Let* $\ell : \Theta \times \mathcal{Z} \to \mathbb{R}$ *and* $c : \mathcal{Z} \times \mathcal{Z} \to \mathbb{R}_+$ *be continuous. Let* $\phi_\gamma(\theta; z_0) = \sup_{z \in \mathcal{Z}} \{\ell(\theta; z) - \gamma c(z, z_0)\}$ *be the robust surrogate* (2b). *For any distribution* $Q$ *and any* $\rho > 0$,

$$\sup_{P : W_c(P,Q) \leq \rho} \mathbb{E}_P[\ell(\theta; Z)] = \inf_{\gamma \geq 0} \{\gamma \rho + \mathbb{E}_Q[\phi_\gamma(\theta; Z)]\}, \tag{5}$$

*and for any* $\gamma \geq 0$*, we have*

$$\sup_P \{\mathbb{E}_P[\ell(\theta; Z)] - \gamma W_c(P, Q)\} = \mathbb{E}_Q[\phi_\gamma(\theta; Z)]. \tag{6}$$

Leveraging the insight (4), we give up the requirement that we wish a prescribed amount $\rho$ of robustness (solving the worst-case problem (1) for $\mathcal{P} = \{P : W_c(P, P_0) \leq \rho\}$) and focus instead on the Lagrangian penalty problem (2) and its empirical counterpart

$$\underset{\theta \in \Theta}{\text{minimize}} \left\{ F_n(\theta) := \sup_P \left\{ \mathbb{E}[\ell(\theta; Z)] - \gamma W_c(P, \widehat{P}_n) \right\} = \mathbb{E}_{\widehat{P}_n}[\phi_\gamma(\theta; Z)] \right\}. \tag{7}$$

## 2.1 Optimizing the robust loss by stochastic gradient descent

We now develop stochastic gradient-type methods for the relaxed robust problem (7), making clear the computational benefits of relaxing the strict robustness requirements of formulation (5). We begin with assumptions we require, which roughly quantify the amount of robustness we can provide.

**Assumption A.** *The function* $c : \mathcal{Z} \times \mathcal{Z} \to \mathbb{R}_+$ *is continuous. For each* $z_0 \in \mathcal{Z}$*,* $c(\cdot, z_0)$ *is 1-strongly convex with respect to the norm* $\|\cdot\|$.

To guarantee that the robust surrogate (2b) is tractably computable, we also require a few smoothness assumptions. Let $\|\cdot\|_*$ be the dual norm to $\|\cdot\|$; we abuse notation by using the same norm $\|\cdot\|$ on $\Theta$ and $\mathcal{Z}$, though the specific norm is clear from context.

**Assumption B.** *The loss* $\ell : \Theta \times \mathcal{Z} \to \mathbb{R}$ *satisfies the Lipschitzian smoothness conditions*

$$\|\nabla_\theta \ell(\theta; z) - \nabla_\theta \ell(\theta'; z)\|_* \leq L_{\theta\theta} \|\theta - \theta'\|, \quad \|\nabla_z \ell(\theta; z) - \nabla_z \ell(\theta; z')\|_* \leq L_{zz} \|z - z'\|,$$
$$\|\nabla_\theta \ell(\theta; z) - \nabla_\theta \ell(\theta; z')\|_* \leq L_{\theta z} \|z - z'\|, \quad \|\nabla_z \ell(\theta; z) - \nabla_z \ell(\theta'; z)\|_* \leq L_{z\theta} \|\theta - \theta'\|.$$

These properties guarantee both (i) the well-behavedness of the robust surrogate $\phi_\gamma$ and (ii) its efficient computability. Making point (i) precise, Lemma 1 shows that if $\gamma$ is large enough and Assumption B holds, the surrogate $\phi_\gamma$ is still smooth. Throughout, we assume $\Theta \subseteq \mathbb{R}^d$.

**Lemma 1.** *Let* $f : \Theta \times \mathcal{Z} \to \mathbb{R}$ *be differentiable and* $\lambda$*-strongly concave in* $z$ *with respect to the norm* $\|\cdot\|$*, and define* $\bar{f}(\theta) = \sup_{z \in \mathcal{Z}} f(\theta, z)$*. Let* $g_\theta(\theta, z) = \nabla_\theta f(\theta, z)$ *and* $g_z(\theta, z) = \nabla_z f(\theta, z)$*, and assume* $g_\theta$ *and* $g_z$ *satisfy Assumption B with* $\ell(\theta; z)$ *replaced with* $f(\theta, z)$*. Then* $\bar{f}$ *is differentiable, and letting* $z^\star(\theta) = \operatorname{argmax}_{z \in \mathcal{Z}} f(\theta, z)$*, we have* $\nabla \bar{f}(\theta) = g_\theta(\theta, z^\star(\theta))$*. Moreover,*

$$\|z^\star(\theta_1) - z^\star(\theta_2)\| \leq \frac{L_{z\theta}}{\lambda} \|\theta_1 - \theta_2\| \quad and \quad \|\nabla \bar{f}(\theta) - \nabla \bar{f}(\theta')\|_\star \leq \left(L_{\theta\theta} + \frac{L_{\theta z} L_{z\theta}}{\lambda}\right) \|\theta - \theta'\|.$$

See Section C.2 for the proof. Fix $z_0 \in \mathcal{Z}$ and focus on the $\ell_2$-norm case where $c(z, z_0)$ satisfies Assumption A with $\|\cdot\|_2$. Noting that $f(\theta, z) := \ell(\theta, z) - \gamma c(z, z_0)$ is $(\gamma - L_{zz})$-strongly concave from the insight (4) (with $L := L_{zz}$), let us apply Lemma 1. Under Assumptions A, B, $\phi_\gamma(\cdot; z_0)$ then has $L = L_{\theta\theta} + \frac{L_{\theta z} L_{z\theta}}{[\gamma - L_{zz}]_+}$-Lipschitz gradients, and

$$\nabla_\theta \phi_\gamma(\theta; z_0) = \nabla_\theta \ell(\theta; z^\star(z_0, \theta)) \quad \text{where} \quad z^\star(z_0, \theta) = \operatorname*{argmax}_{z \in \mathcal{Z}} \{\ell(\theta; z) - \gamma c(z, z_0)\}.$$

---

**Algorithm 1** Distributionally robust optimization with adversarial training

---

INPUT: Sampling distribution $P_0$, constraint sets $\Theta$ and $\mathcal{Z}$, stepsize sequence $\{\alpha_t > 0\}_{t=0}^{T-1}$
**for** $t = 0, \ldots, T - 1$ **do**
    Sample $z^t \sim P_0$ and find an $\epsilon$-approximate maximizer $\hat{z}^t$ of $\ell(\theta^t; z) - \gamma c(z, z^t)$
    $\theta^{t+1} \leftarrow \text{Proj}_\Theta(\theta^t - \alpha_t \nabla_\theta \ell(\theta^t; \hat{z}^t))$

---

This motivates Algorithm 1, a stochastic-gradient approach for the penalty problem (7). The benefits of Lagrangian relaxation become clear here: for $\ell(\theta; z)$ smooth in $z$ and $\gamma$ large enough, gradient ascent on $\ell(\theta^t; z) - \gamma c(z, z^t)$ in $z$ converges linearly and we can compute (approximate) $\hat{z}^t$ efficiently (we initialize our inner gradient ascent iterations with the sampled natural example $z^t$).

Convergence properties of Algorithm 1 depend on the loss $\ell$. When $\ell$ is convex in $\theta$ and $\gamma$ is large enough that $z \mapsto (\ell(\theta; z) - \gamma c(z, z_0))$ is concave for all $(\theta, z_0) \in \Theta \times \mathcal{Z}$, we have a stochastic monotone variational inequality, which is efficiently solvable (Juditsky et al., 2011; Chen et al., 2014) with convergence rate $1/\sqrt{T}$. When the loss $\ell$ is nonconvex in $\theta$, the following theorem guarantees convergence to a stationary point of problem (7) at the same rate when $\gamma \geq L_{\mathsf{zz}}$. Recall that $F(\theta) = \mathbb{E}_{P_0}[\phi_\gamma(\theta; Z)]$ is the robust surrogate objective for the Lagrangian relaxation (2).

**Theorem 2** (Convergence of Nonconvex SGD). *Let Assumptions A and B hold with the $\ell_2$-norm and let $\Theta = \mathbb{R}^d$. Let $\Delta_F \geq F(\theta^0) - \inf_\theta F(\theta)$. Assume $\mathbb{E}[\|\nabla F(\theta) - \nabla_\theta \phi_\gamma(\theta, Z)\|_2^2] \leq \sigma^2$ and take constant stepsizes $\alpha = \sqrt{\frac{\Delta_F}{L_\phi T \sigma^2}}$ where $L_\phi := L_{\theta\theta} + \frac{L_{\theta z} L_{z\theta}}{\gamma - L_{\mathsf{zz}}}$. For $T \geq \frac{L_\phi \Delta_F}{\sigma^2}$, Algorithm 1 satisfies*

$$\frac{1}{T} \sum_{t=0}^{T-1} \mathbb{E}\left[\left\|\nabla F(\theta^t)\right\|_2^2\right] - \frac{4L_{\theta z}^2}{\gamma - L_{\mathsf{zz}}}\epsilon \leq 4\sigma\sqrt{\frac{L_\phi \Delta_F}{T}}.$$

See Section C.3 for the proof. We make a few remarks. First, the condition $\mathbb{E}[\|\nabla F(\theta) - \nabla_\theta \phi_\gamma(\theta, Z)\|_2^2] \leq \sigma^2$ holds (to within a constant factor) whenever $\|\nabla_\theta \ell(\theta, z)\|_2 \leq \sigma$ for all $\theta, z$. Theorem 2 shows that the stochastic gradient method achieves the rates of convergence on the penalty problem (7) achievable in standard smooth non-convex optimization (Ghadimi & Lan, 2013). The accuracy parameter $\epsilon$ has a *fixed* effect on optimization accuracy, independent of $T$: approximate maximization has limited effects.

Key to the convergence guarantee of Theorem 2 is that the loss $\ell$ is smooth in $z$: the inner supremum (2b) is NP-hard to compute for non-smooth deep networks (see Lemma 2 in Section B for a proof of this for ReLU's). The smoothness of $\ell$ is essential so that a penalized version $\ell(\theta, z) - \gamma c(z, z_0)$ is concave in $z$ (which can be approximately verified by computing Hessians $\nabla_{zz}^2 \ell(\theta, z)$ for each training datapoint), allowing computation and our coming certificates of optimality. Replacing ReLU's with sigmoids or ELU's (Clevert et al., 2015) allows us to apply Theorem 2, making distributionally robust optimization tractable for deep learning.

In supervised-learning scenarios, we are often interested in adversarial perturbations only to feature vectors (and not labels). Letting $Z = (X, Y)$ where $X$ denotes the feature vector (covariates) and $Y$ the label, this is equivalent to defining the Wasserstein cost function $c : \mathcal{Z} \times \mathcal{Z} \to \mathbb{R}_+ \cup \{\infty\}$ by

$$c(z, z') := c_x(x, x') + \infty \cdot \mathbf{1}\{y \neq y'\} \tag{8}$$

where $c_x : \mathcal{X} \times \mathcal{X} \to \mathbb{R}_+$ is the transportation cost for the feature vector $X$. All of our results suitably generalize to this setting with minor modifications to the robust surrogate (2b) and the above assumptions (see Section D). Similarly, our distributionally robust framework (2) is general enough to consider adversarial perturbations to only an arbitrary subset of coordinates in $Z$. For example, it may be appropriate in certain applications to hedge against adversarial perturbations to a small fixed region of an image (Brown et al., 2017). By suitably modifying the cost function $c(z, z')$ to take value $\infty$ outside this small region, our general formulation covers such variants.

## 3 CERTIFICATE OF ROBUSTNESS AND GENERALIZATION

From results in the previous section, Algorithm 1 provably learns to protect against adversarial perturbations of the form (7) on the training dataset. Now we show that such procedures generalize,

allowing us to prevent attacks on the test set. Our subsequent results hold uniformly over the space of parameters $\theta \in \Theta$, including $\theta_{\mathrm{WRM}}$, the output of the stochastic gradient descent procedure in Section 2.1. Our first main result, presented in Section 3.1, gives a data-dependent upper bound on the population worst-case objective $\sup_{P:W_c(P,P_0)\leq\rho} \mathbb{E}_P[\ell(\theta;Z)]$ for any arbitrary level of robustness $\rho$; this bound is optimal for $\rho = \widehat{\rho}_n$, the level of robustness achieved for the empirical distribution by solving (7). Our bound is efficiently computable and hence *certifies* a level of robustness for the worst-case population objective. Second, we show in Section 3.2 that adversarial perturbations on the training set (in a sense) generalize: solving the empirical penalty problem (7) guarantees a similar level of robustness as directly solving its population counterpart (2).

## 3.1 Certificate of robustness

Our main result in this section is a data-dependent upper bound for the worst-case population objective: $\sup_{P:W_c(P,P_0)\leq\rho} \mathbb{E}_P[\ell(\theta;Z)] \leq \gamma\rho + \mathbb{E}_{\widehat{P}_n}[\phi_\gamma(\theta;Z)] + O(1/\sqrt{n})$ for all $\theta \in \Theta$, with high probability. To make this rigorous, fix $\gamma > 0$, and consider the worst-case perturbation, typically called the *transportation map* or Monge map (Villani, 2009),

$$T_\gamma(\theta;z_0) := \underset{z\in\mathcal{Z}}{\operatorname{argmax}}\{\ell(\theta;z) - \gamma c(z,z_0)\}. \tag{9}$$

Under our assumptions, $T_\gamma$ is easily computable when $\gamma \geq L_{\mathsf{zz}}$. Letting $\delta_z$ denote the point mass at $z$, Proposition 1 shows the empirical maximizers of the Lagrangian formulation (6) are attained by

$$P_n^*(\theta) := \underset{P}{\operatorname{argmax}}\left\{\mathbb{E}_P[\ell(\theta;Z)] - \gamma W_c(P,\widehat{P}_n)\right\} = \frac{1}{n}\sum_{i=1}^n \delta_{T_\gamma(\theta,Z_i)} \quad \text{and}$$

$$\widehat{\rho}_n(\theta) := W_c(P_n^*(\theta),\widehat{P}_n) = \mathbb{E}_{\widehat{P}_n}[c(T_\gamma(\theta;Z),Z)]. \tag{10}$$

Our results imply, in particular, that the empirical worst-case loss $\mathbb{E}_{P_n^*}[\ell(\theta;Z)]$ gives a *certificate of robustness* to (population) Wasserstein perturbations up to level $\widehat{\rho}_n$. $\mathbb{E}_{P_n^*(\theta)}[\ell(\theta;Z)]$ is efficiently computable via (10), providing a data-dependent guarantee for the worst-case population loss.

Our bound relies on the usual covering numbers for the model class $\{\ell(\theta;\cdot) : \theta \in \Theta\}$ as the notion of complexity (e.g. van der Vaart & Wellner, 1996), so, despite the infinite-dimensional problem (7), we retain the same uniform convergence guarantees typical of empirical risk minimization. Recall that for a set $V$, a collection $v_1,\ldots,v_N$ is an $\epsilon$-*cover* of $V$ in norm $\|\cdot\|$ if for each $v \in \mathcal{V}$, there exists $v_i$ such that $\|v - v_i\| \leq \epsilon$. The *covering number* of $V$ with respect to $\|\cdot\|$ is

$$N(V,\epsilon,\|\cdot\|) := \inf\{N \in \mathbb{N} \mid \text{there is an } \epsilon\text{-cover of } V \text{ with respect to } \|\cdot\|\}.$$

For $\mathcal{F} := \{\ell(\theta,\cdot) : \theta \in \Theta\}$ equipped with the $L^\infty(\mathcal{Z})$ norm $\|f\|_{L^\infty(\mathcal{Z})} := \sup_{z\in\mathcal{Z}}|f(z)|$, we state our results in terms of $\|\cdot\|_{L^\infty(\mathcal{Z})}$-covering numbers of $\mathcal{F}$. To ease notation, we let

$$\epsilon_n(t) := \gamma b_1 \sqrt{\frac{M_\ell}{n}} \int_0^1 \sqrt{\log N(\mathcal{F}, M_\ell\epsilon, \|\cdot\|_{L^\infty(\mathcal{Z})})}d\epsilon + b_2 M_\ell\sqrt{\frac{t}{n}}$$

where $b_1, b_2$ are numerical constants.

We are now ready to state the main result of this section. We first show from the duality result (6) that we can provide an upper bound for the worst-case population performance for any level of robustness $\rho$. For $\rho = \widehat{\rho}_n(\theta)$ and $\theta = \theta_{\mathrm{WRM}}$, this certificate is (in a sense) tight as we see below.

**Theorem 3.** *Assume* $|\ell(\theta;z)| \leq M_\ell$ *for all* $\theta \in \Theta$ *and* $z \in \mathcal{Z}$. *Then, for a fixed* $t > 0$ *and numerical constants* $b_1, b_2 > 0$, *with probability at least* $1 - e^{-t}$, *simultaneously for all* $\theta \in \Theta$, $\rho \geq 0$, $\gamma \geq 0$,

$$\sup_{P:W_c(P,P_0)\leq\rho} \mathbb{E}_P[\ell(\theta;Z)] \leq \gamma\rho + \mathbb{E}_{\widehat{P}_n}[\phi_\gamma(\theta;Z)] + \epsilon_n(t). \tag{11}$$

*In particular, if* $\rho = \widehat{\rho}_n(\theta)$ *then with probability at least* $1 - e^{-t}$, *for all* $\theta \in \Theta$

$$\sup_{P:W_c(P,P_0)\leq\widehat{\rho}_n(\theta)} \mathbb{E}_P[\ell(\theta;Z)] \leq \gamma\widehat{\rho}_n(\theta) + \mathbb{E}_{\widehat{P}_n}[\phi_\gamma(\theta;Z)] + \epsilon_n(t)$$

$$= \sup_{P:W_c(P,\widehat{P}_n)\leq\widehat{\rho}_n(\theta)} \mathbb{E}_P[\ell(\theta;Z)] + \epsilon_n(t). \tag{12}$$

See Section C.4 for its proof. We now give a concrete variant of Theorem 3 for Lipschitz functions. When $\Theta$ is finite-dimensional ($\Theta \subset \mathbb{R}^d$), Theorem 3 provides a robustness guarantee scaling linearly with $d$ despite the infinite-dimensional Wasserstein penalty. Assuming there exist $\theta_0 \in \Theta$, $M_{\theta_0} < \infty$ such that $|\ell(\theta_0; z)| \leq M_{\theta_0}$ for all $z \in \mathcal{Z}$, we have the following corollary (see proof in Section C.5).

**Corollary 1.** *Let $\ell(\cdot; z)$ be L-Lipschitz with respect to some norm $\|\cdot\|$ for all $z \in \mathcal{Z}$. Assume that $\Theta \subset \mathbb{R}^d$ satisfies $\mathrm{diam}(\Theta) = \sup_{\theta, \theta' \in \Theta} \|\theta - \theta'\| < \infty$. Then, the bounds (11) and (12) hold with*

$$\epsilon_n(t) = b_1 \sqrt{\frac{d(L\,\mathrm{diam}(\Theta) + M_{\theta_0})}{n}} + b_2 (L\,\mathrm{diam}(\Theta) + M_{\theta_0}) \sqrt{\frac{t}{n}} \tag{13}$$

*for some numerical constants $b_1, b_2 > 0$.*

A key consequence of the bound (11) is that $\gamma\rho + \mathbb{E}_{\widehat{P}_n}[\phi_\gamma(\theta; Z)]$ *certifies* robustness for the worst-case population objective for any $\rho$ and $\theta$. For a given $\theta$, this certificate is tightest at the achieved level of robustness $\widehat{\rho}_n(\theta)$, as noted in the refined bound (12) which follows from the duality result

$$\underbrace{\mathbb{E}_{\widehat{P}_n}[\phi_\gamma(\theta; Z)]}_{\text{surrogate loss}} + \underbrace{\gamma\widehat{\rho}_n(\theta)}_{\text{robustness}} = \sup_{P: W_c(P, \widehat{P}_n) \leq \widehat{\rho}_n(\theta)} \mathbb{E}_P[\ell(\theta; Z)] = \mathbb{E}_{P_n^*(\theta)}[\ell(\theta; Z)]. \tag{14}$$

(See Section C.4 for a proof of these equalities.) We expect $\theta_{\mathrm{WRM}}$, the output of Algorithm 1, to be close to the minimizer of the surrogate loss $\mathbb{E}_{\widehat{P}_n}[\phi_\gamma(\theta; Z)]$ and therefore have the best guarantees. Most importantly, the certificate (14) is easy to compute via expression (10): as noted in Section 2.1, the mappings $T(\theta, Z_i)$ are efficiently computable for large enough $\gamma$, and $\widehat{\rho}_n = \mathbb{E}_{\widehat{P}_n}[c(T(\theta, Z), Z)]$.

The bounds (11)–(13) may be too large—because of their dependence on covering numbers and dimension—for practical use in security-critical applications. With that said, the strong duality result, Proposition 1, still applies to any distribution. In particular, given a collection of test examples $Z_i^{\mathrm{test}}$, we may interrogate possible losses under perturbations for the *test* examples by noting that, if $\widehat{P}_{\mathrm{test}}$ denotes the empirical distribution on the test set (say, with putative assigned labels), then

$$\frac{1}{n_{\mathrm{test}}} \sum_{i=1}^{n} \sup_{z: c(z, Z_i^{\mathrm{test}}) \leq \rho} \{\ell(\theta; z)\} \leq \sup_{P: W_c(P, \widehat{P}_{\mathrm{test}}) \leq \rho} \mathbb{E}_P[\ell(\theta; Z)] \leq \gamma\rho + \mathbb{E}_{\widehat{P}_{\mathrm{test}}}[\phi_\gamma(\theta; Z)] \tag{15}$$

for all $\gamma, \rho \geq 0$. Whenever $\gamma$ is large enough (so that this is tight for small $\rho$), we may efficiently compute the Monge-map (9) and the test loss (15) to guarantee bounds on the sensitivity of a parameter $\theta$ to a particular sample and predicted labeling based on the sample.

## 3.2 GENERALIZATION OF ADVERSARIAL EXAMPLES

We can also show that the level of robustness on the training set generalizes. Our starting point is Lemma 1, which shows that $T_\gamma(\cdot; z)$ is smooth under Assumptions A and B:

$$\|T_\gamma(\theta_1; z) - T_\gamma(\theta_2; z)\| \leq \frac{L_{\mathsf{z}\theta}}{[\gamma - L_{\mathsf{zz}}]_+} \|\theta_1 - \theta_2\| \tag{16}$$

for all $\theta_1, \theta_2$, where we recall that $L_{\mathsf{zz}}$ is the Lipschitz constant of $\nabla_z \ell(\theta; z)$. Leveraging this smoothness, we show that $\widehat{\rho}_n(\theta) = \mathbb{E}_{\widehat{P}_n}[c(T_\gamma(\theta; Z), Z)]$, the level of robustness achieved for the empirical problem, concentrates uniformly around its population counterpart.

**Theorem 4.** *Let $\mathcal{Z} \subset \{z \in \mathbb{R}^m : \|z\| \leq M_{\mathsf{z}}\}$ so that $\|Z\| \leq M_{\mathsf{z}}$ almost surely and assume either that (i) $c(\cdot, \cdot)$ is $L_{\mathsf{c}}$-Lipschitz over $\mathcal{Z}$ with respect to the norm $\|\cdot\|$ in each argument, or (ii) that $\ell(\theta, z) \in [0, M_\ell]$ and $z \mapsto \ell(\theta, z)$ is $\gamma L_{\mathsf{c}}$-Lipschitz for all $\theta \in \Theta$.*

*If Assumptions A and B hold, then with probability at least $1 - e^{-t}$,*

$$\sup_{\theta \in \Theta} |\mathbb{E}_{\widehat{P}_n}[c(T_\gamma(\theta; Z), Z)] - \mathbb{E}_{P_0}[c(T_\gamma(\theta; Z), Z)]| \leq 4B \sqrt{\frac{1}{n} \left( t + \log N \left( \Theta, \frac{[\gamma - L_{\mathsf{zz}}]_+ t}{4 L_{\mathsf{c}} L_{\mathsf{z}\theta}}, \|\cdot\| \right) \right)}. \tag{17}$$

*where $B = L_{\mathsf{c}} M_{\mathsf{z}}$ under assumption (i) and $B = M_\ell / \gamma$ under assumption (ii).*

See Section C.6 for the proof. For $\Theta \subset \mathbb{R}^d$, we have $\log N(\Theta, \epsilon, \|\cdot\|) \leq d \log(1 + \frac{\text{diam}(\Theta)}{\epsilon})$ so that the bound (30) gives the usual $\sqrt{d/n}$ generalization rate for the distance between adversarial perturbations and natural examples. Another consequence of Theorem 4 is that $\widehat{\rho}_n(\theta_{\text{WRM}})$ in the certificate (12) is positive as long as the loss $\ell$ is not completely invariant to data. To see this, note from the optimality conditions for $T_\gamma(\theta; Z)$ that $\mathbb{E}_{P_0}[c(T_\gamma(\theta; Z), Z)] = 0$ iff $\nabla_z \ell(\theta; z) = 0$ almost surely, and hence for large enough $n$, we have $\widehat{\rho}_n(\theta) > 0$ by the bound (30).

# 4 EXPERIMENTS

Our technique for distributionally robust optimization with adversarial training extends beyond supervised learning. To that end, we present empirical evaluations on supervised and reinforcement learning tasks where we compare performance with empirical risk minimization (ERM) and, where appropriate, models trained with the fast-gradient method (3) (FGM) (Goodfellow et al., 2015), its iterated variant (IFGM) (Kurakin et al., 2016), and the projected-gradient method (PGM) (Madry et al., 2017). PGM augments stochastic gradient steps for the parameter $\theta$ with projected gradient ascent over $x \mapsto \ell(\theta; x, y)$, iterating (for data point $x_i, y_i$)

$$\Delta x_i^{t+1}(\theta) := \operatorname*{argmax}_{\|\eta\|_p \leq \epsilon}\{\nabla_x \ell(\theta; x_i^t, y_i)^T \eta\} \text{ and } x_i^{t+1} := \Pi_{\mathcal{B}_{\epsilon,p}(x_i^t)}\left\{x_i^t + \alpha_t \Delta x_i^t(\theta)\right\} \qquad (18)$$

for $t = 1, \ldots, T_{\text{adv}}$, where $\Pi$ denotes projection onto $\mathcal{B}_{\epsilon,p}(x_i) := \{x : \|x - x_i\|_p \leq \epsilon\}$.

The adversarial training literature (e.g. Goodfellow et al. (2015)) usually considers $\|\cdot\|_\infty$-norm attacks, which allow imperceptible perturbations to all input features. In most scenarios, however, it is reasonable to defend against weaker adversaries that instead perturb influential features more. We consider this setting and train against $\|\cdot\|_2$-norm attacks. Namely, we use the squared Euclidean cost for the feature vectors $c_x(x, x') := \|x - x'\|_2^2$ and define the overall cost as the covariate-shift adversary (8) for WRM (Algorithm 1), and we use $p = 2$ for FGM, IFGM, PGM training in all experiments; we still test against adversarial perturbations with respect to the norms $p = 2, \infty$. We use $T_{\text{adv}} = 15$ iterations for all iterative methods (IFGM, PGM, and WRM) in training and attacks.

In Section 4.1, we visualize differences between our approach and ad-hoc methods to illustrate the benefits of certified robustness. In Section 4.2 we consider a supervised learning problem for MNIST where we adversarially perturb the test data. Finally, we consider a reinforcement learning problem in Section 4.3, where the Markov decision process used for training differs from that for testing.

WRM enjoys the theoretical guarantees of Sections 2 and 3 for large $\gamma$, but for small $\gamma$ (large adversarial budgets), WRM becomes a heuristic like other methods. In Appendix A.4, we compare WRM with other methods on attacks with large adversarial budgets. In Appendix A.5, we further compare WRM—which is trained to defend against $\|\cdot\|_2$-adversaries—with other heuristics trained to defend against $\|\cdot\|_\infty$-adversaries. WRM matches or outperforms other heuristics against imperceptible attacks, while it underperforms for attacks with large adversarial budgets.

## 4.1 VISUALIZING THE BENEFITS OF CERTIFIED ROBUSTNESS

For our first experiment, we generate synthetic data $Z = (X, Y) \sim P_0$ by $X_i \overset{\text{iid}}{\sim} \mathsf{N}(0_2, I_2)$ with labels $Y_i = \text{sign}(\|x\|_2 - \sqrt{2})$, where $X \in \mathbb{R}^2$ and $I_2$ is the identity matrix in $\mathbb{R}^2$. Furthermore, to create a wide margin separating the classes, we remove data with $\|X\|_2 \in (\sqrt{2}/1.3, 1.3\sqrt{2})$. We train a small neural network with 2 hidden layers of size 4 and 2 and either all ReLU or all ELU activations between layers, comparing our approach (WRM) with ERM and the 2-norm FGM. For our approach we use $\gamma = 2$, and to make fair comparisons with FGM we use

$$\epsilon^2 = \widehat{\rho}_n(\theta_{\text{WRM}}) = W_c(P_n^*(\theta_{\text{WRM}}), \widehat{P}_n) = \mathbb{E}_{\widehat{P}_n}[c(T(\theta_{\text{WRM}}, Z), Z)], \qquad (19)$$

for the fast-gradient perturbation magnitude $\epsilon$, where $\theta_{\text{WRM}}$ is the output of Algorithm 1.[1]

Figure 1 illustrates the classification boundaries for the three training procedures over the ReLU-activated (Figure 1(a)) and ELU-activated (Figure 1(b)) models. Since 70% of the data are of the

---

[1] For ELU activations with scale parameter 1, $\gamma = 2$ makes problem (2b) strongly concave over the training data. ReLU's have no guarantees, but we use 15 gradient steps with stepsize $1/\sqrt{t}$ for both activations.

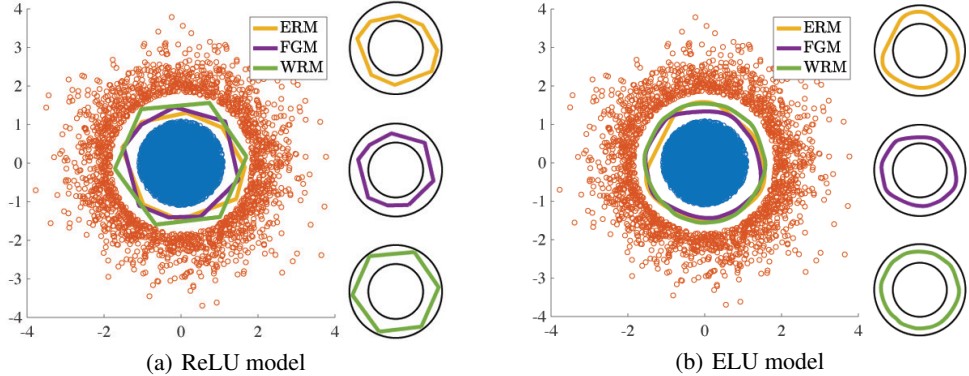

**Figure 1.** Experimental results on synthetic data. Training data are shown in blue and red. Classification boundaries are shown in yellow, purple, and green for ERM, FGM, and WRM respectively. The boundaries are shown with the training data as well as separately with the true class boundaries.

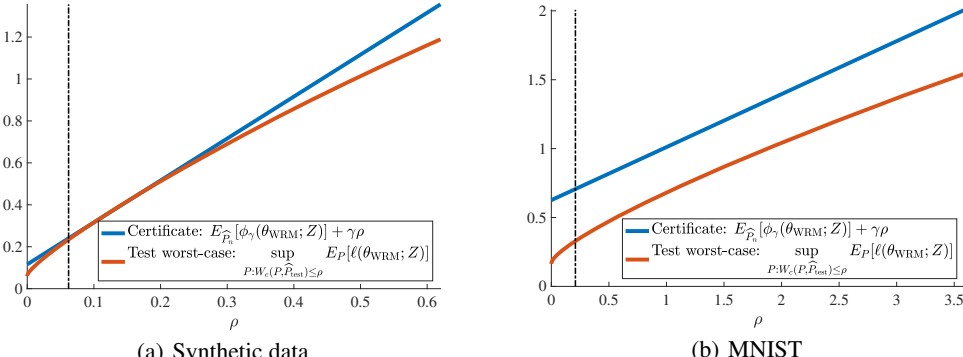

**Figure 2.** Empirical comparison between certificate of robustness (11) (blue) and test worst-case performance (red) for experiments with (a) synthetic data and (b) MNIST. We omit the certificate's error term $\epsilon_n(t)$. The vertical bar indicates the achieved level of robustness on the training set $\widehat{\rho}_n(\theta_{\mathrm{WRM}})$.

blue class ($\|X\|_2 \leq \sqrt{2}/1.3$), distributional robustness favors pushing the classification boundary outwards; intuitively, adversarial examples are most likely to come from pushing blue points outwards across the boundary. ERM and FGM suffer from sensitivities to various regions of the data, as evidenced by the lack of symmetry in their classification boundaries. For both activations, WRM pushes the classification boundaries further outwards than ERM or FGM. However, WRM with ReLU's still suffers from sensitivities (e.g. radial asymmetry in the classification surface) due to the lack of robustness guarantees. WRM with ELU's provides a certified level of robustness, yielding an axisymmetric classification boundary that hedges against adversarial perturbations in all directions.

Recall that our *certificates of robustness* on the worst-case performance given in Theorem 3 applies for any level of robustness $\rho$. In Figure 2(a), we plot our certificate (11) against the out-of-sample (test) worst-case performance $\sup_{P:W_c(P,P_0)\leq\rho} \mathbb{E}_P[\ell(\theta;Z)]$ for WRM with ELU's. Since the worst-case loss is hard to evaluate directly, we solve its Lagrangian relaxation (6) for different values of $\gamma_{\mathrm{adv}}$. For each $\gamma_{\mathrm{adv}}$, we consider the distance to adversarial examples in the test dataset

$$\widehat{\rho}_{\mathrm{test}}(\theta) := \mathbb{E}_{\widehat{P}_{\mathrm{test}}}[c(T_{\gamma_{\mathrm{adv}}}(\theta, Z), Z)], \tag{20}$$

where $\widehat{P}_{\mathrm{test}}$ is the test distribution, $c(z, z') := \|x - x'\|_2^2 + \infty \cdot \mathbf{1}\{y \neq y'\}$ as before, and $T_{\gamma_{\mathrm{adv}}}(\theta, Z) = \arg\max_z\{\ell(\theta; z) - \gamma_{\mathrm{adv}}c(z, Z)\}$ is the adversarial perturbation of $Z$ (Monge map) for the model $\theta$. The worst-case losses on the test dataset are then given by

$$\mathbb{E}_{\widehat{P}_{\mathrm{test}}}[\phi_{\gamma_{\mathrm{adv}}}(\theta_{\mathrm{WRM}}; Z)] + \gamma_{\mathrm{adv}}\widehat{\rho}_{\mathrm{test}}(\theta_{\mathrm{WRM}}) = \sup_{P:W_c(P,P_{\mathrm{test}})\leq\widehat{\rho}_{\mathrm{test}}(\theta_{\mathrm{WRM}})} \mathbb{E}_P[\ell(\theta_{\mathrm{WRM}}; Z)].$$

As anticipated, our certificate is almost tight near the achieved level of robustness $\widehat{\rho}_n(\theta_{\mathrm{WRM}})$ for WRM (10) and provides a performance guarantee even for other values of $\rho$.

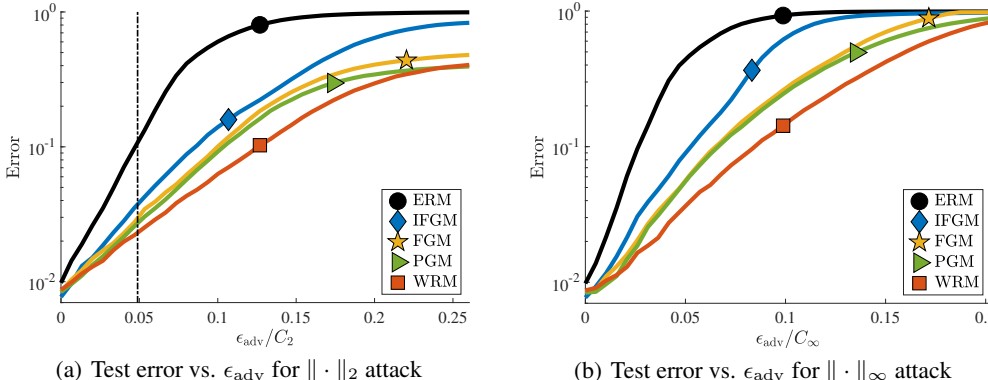

(a) Test error vs. $\epsilon_{\text{adv}}$ for $\|\cdot\|_2$ attack    (b) Test error vs. $\epsilon_{\text{adv}}$ for $\|\cdot\|_\infty$ attack

**Figure 3.** PGM attacks on the MNIST dataset. (a) and (b) show test misclassification error vs. the adversarial perturbation level $\epsilon_{\text{adv}}$ for the PGM attack with respect to Euclidean and $\infty$ norms respectively. The vertical bar in (a) indicates the perturbation level used for training the PGM, FGM, and IFGM models as well as the estimated radius $\sqrt{\widehat{\rho}_n(\theta_{\text{WRM}})}$. For MNIST, $C_2 = 9.21$ and $C_\infty = 1.00$.

## 4.2    LEARNING A MORE ROBUST CLASSIFIER

We now consider a standard benchmark—training a neural network classifier on the MNIST dataset. The network consists of $8 \times 8$, $6 \times 6$, and $5 \times 5$ convolutional filter layers with ELU activations followed by a fully connected layer and softmax output. We train WRM with $\gamma = 0.04 \mathbb{E}_{\widehat{P}_n}[\|X\|_2]$, and for the other methods we choose $\epsilon$ as the level of robustness achieved by WRM (19).[2] In the figures, we scale the budgets $1/\gamma_{\text{adv}}$ and $\epsilon_{\text{adv}}$ for the adversary with $C_p := \mathbb{E}_{\widehat{P}_n}[\|X\|_p]$.[3]

First, in Figure 2(b) we again illustrate the validity of our certificate of robustness (11) for the worst-case test performance for arbitrary level of robustness $\rho$. We see that our certificate provides a performance guarantee for out-of-sample worst-case performance.

We now compare adversarial training techniques. All methods achieve at least 99% test-set accuracy, implying there is little test-time penalty for the robustness levels ($\epsilon$ and $\gamma$) used for training. It is thus important to distinguish the methods' abilities to combat attacks. We test performance of the five methods (ERM, FGM, IFGM, PGM, WRM) under PGM attacks (18) with respect to 2- and $\infty$-norms. In Figure 3(a) and (b), all adversarial methods outperform ERM, and WRM offers more robustness even with respect to these PGM attacks. Training with the Euclidean cost still provides robustness to $\infty$-norm fast gradient attacks. We provide further evidence in Appendix A.1.

Next we study stability of the loss surface with respect to perturbations to inputs. We note that small values of $\widehat{\rho}_{\text{test}}(\theta)$, the distance to adversarial examples (20), correspond to small magnitudes of $\nabla_z \ell(\theta; z)$ in a neighborhood of the nominal input, which ensures stability of the model. Figure 4(a) shows that $\widehat{\rho}_{\text{test}}$ differs by orders of magnitude between the training methods (models $\theta = \theta_{\text{ERM}}, \theta_{\text{FGM}}, \theta_{\text{IFGM}}, \theta_{\text{PGM}}, \theta_{\text{WRM}}$); the trend is nearly uniform over all $\gamma_{\text{adv}}$, with $\theta_{\text{WRM}}$ being the most stable. Thus, we see that our adversarial-training method defends against gradient-exploiting attacks by reducing the magnitudes of gradients near the nominal input.

In Figure 4(b) we provide a qualitative picture by adversarially perturbing a single test datapoint until the model misclassifies it. Specifically, we again consider WRM attacks and we decrease $\gamma_{\text{adv}}$ until each model misclassifies the input. The original label is 8, whereas on the adversarial examples IFGM predicts 2, PGM predicts 0, and the other models predict 3. WRM's "misclassifications" appear consistently reasonable to the human eye (see Appendix A.2 for examples of other digits); WRM defends against gradient-based exploits by learning a representation that makes gradients point towards inputs of other classes. Together, Figures 4(a) and (b) depict our method's defense mechanisms to gradient-based attacks: creating a more stable loss surface by reducing the magnitude of gradients and improving their interpretability.

---

[2]For this $\gamma$, $\phi_\gamma(\theta_{\text{WRM}}; z)$ is strongly concave for 98% of the training data.

[3]For the standard MNIST dataset, $C_2 := \mathbb{E}_{\widehat{P}_n} \|X\|_2 = 9.21$ and $C_\infty := \mathbb{E}_{\widehat{P}_n} \|X\|_\infty = 1.00$.

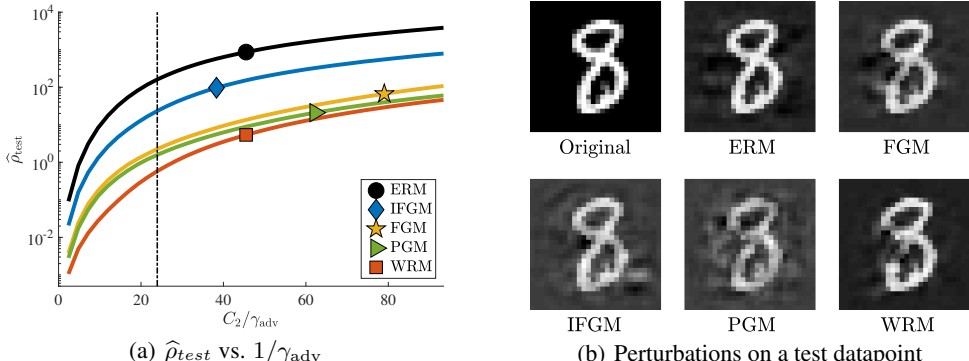

(a) $\widehat{\rho}_{test}$ vs. $1/\gamma_{\mathrm{adv}}$         (b) Perturbations on a test datapoint

**Figure 4.** Stability of the loss surface. In (a), we show the average distance of the perturbed distribution $\widehat{\rho}_{\mathrm{test}}$ for a given $\gamma_{\mathrm{adv}}$, an indicator of local stability to inputs for the decision surface. The vertical bar in (a) indicates the $\gamma$ we use for training WRM. In (b) we visualize the smallest WRM perturbation (largest $\gamma_{\mathrm{adv}}$) necessary to make a model misclassify a datapoint. More examples are in Appendix A.2.

### 4.3 ROBUST MARKOV DECISION PROCESSES

For our final experiments, we consider distributional robustness in the context of Q-learning, a model-free reinforcement learning technique. We consider Markov decision processes (MDP's) $(\mathcal{S}, \mathcal{A}, P_{sa}, r)$ with state space $\mathcal{S}$, action space $\mathcal{A}$, state-action transition probabilities $P_{sa}$, and rewards $r : \mathcal{S} \to \mathbb{R}$. The goal of a reinforcement-learning agent is to maximize (discounted) cumulative rewards $\sum_t \lambda^t \mathbb{E}[r(s^t)]$ (with discount factor $\lambda$); this is analogous to minimizing $\mathbb{E}_P[\ell(\theta; Z)]$ in supervised learning. Robust MDP's consider an ambiguity set $\mathcal{P}_{sa}$ for state-action transitions. The goal is maximizing the worst-case realization $\inf_{P \in \mathcal{P}_{sa}} \sum_t \lambda^t \mathbb{E}_P[r(s^t)]$, analogous to problem (1).

In a standard MDP, Q-learning learns a quality function $Q : \mathcal{S} \times \mathcal{A} \to \mathbb{R}$ via the iterations

$$Q(s^t, a^t) \leftarrow Q(s^t, a^t) + \alpha_t \left( r(s^t) + \lambda \max_a Q(s^{t+1}, a) - Q(s^t, a^t) \right) \qquad (21)$$

such that $\mathrm{argmax}_a Q(s, a)$ is (eventually) the optimal action in state $s$ to maximize cumulative reward. In scenarios where the underlying environment has a continuous state-space and we represent $Q$ with a differentiable function (e.g. Mnih et al. (2015)), we can modify the update (21) with an adversarial state perturbation to incorporate distributional robustness. Namely, we draw the nominal state-transition update $\widehat{s}^{t+1} \sim p_{sa}(s^t, a^t)$, and proceed with the update (21) using the perturbation

$$s^{t+1} \leftarrow \mathrm{argmin}_s \left\{ r(s) + \lambda \max_a Q(s, a) + \gamma c(s, \hat{s}^{t+1}) \right\}. \qquad (22)$$

For large $\gamma$, we can again solve problem (22) efficiently using gradient descent. This procedure provides robustness to uncertainties in state-action transitions. For tabular Q-learning, where we represent $Q$ only over a discretized covering of the underlying state-space, we can either neglect the second term in the update (22) and, after performing the update, round $s^{t+1}$ as usual, or we can perform minimization directly over the discretized covering. In the former case, since the update (22) simply modifies the state-action transitions (independent of $Q$), standard results on convergence for tabular Q-learning (e.g. Szepesvári & Littman (1999)) apply under these adversarial dynamics.

We test our adversarial training procedure in the cart-pole environment, where the goal is to balance a pole on a cart by moving the cart left or right. The environment caps episode lengths to 400 steps and ends the episode prematurely if the pole falls too far from the vertical or the cart translates too far from its origin. We use reward $r(\beta) := e^{-|\beta|}$ for the angle $\beta$ of the pole from the vertical. We use a tabular representation for $Q$ with 30 discretized states for $\beta$ and 15 for its time-derivative $\dot{\beta}$ (we perform the update (22) without the $Q$-dependent term). The action space is binary: push the cart left or right with a fixed force. Due to the nonstationary, policy-dependent radius for the Wasserstein ball, an analogous $\epsilon$ for the fast-gradient method (or other variants) is not well-defined. Thus, we only compare with an agent trained on the nominal MDP. We test both models with perturbations to the physical parameters: we shrink/magnify the pole's mass by 2, the pole's length by 2, and the strength of gravity $g$ by 5. The system's dynamics are such that the heavy, short, and strong-gravity cases are more unstable than the original environment, whereas their counterparts are less unstable.

| Environment | Regular | Robust |
|---|---|---|
| Original | $399.7 \pm 0.1$ | $400.0 \pm 0.0$ |
| Easier environments | | |
| Light | $400.0 \pm 0.0$ | $400.0 \pm 0.0$ |
| Long | $400.0 \pm 0.0$ | $400.0 \pm 0.0$ |
| Soft $g$ | $400.0 \pm 0.0$ | $400.0 \pm 0.0$ |
| Harder environments | | |
| Heavy | $150.1 \pm 4.7$ | $334.0 \pm 3.7$ |
| Short | $245.2 \pm 4.8$ | $400.0 \pm 0.0$ |
| Strong $g$ | $189.8 \pm 2.3$ | $398.5 \pm 0.3$ |

**Table 1.** Episode length over 1000 trials (mean $\pm$ standard error)

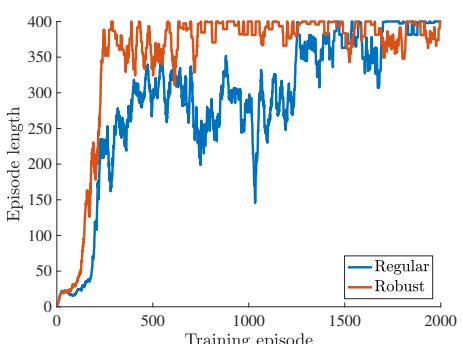

**Figure 5.** Episode lengths during training. The environment caps episodes to 400 steps.

Table 1 shows the performance of the trained models over the original MDP and all of the perturbed MDPs. Both models perform similarly over easier environments, but the robust model greatly outperforms in harder environments. Interestingly, as shown in Figure 5, the robust model also learns more efficiently than the nominal model in the original MDP. We hypothesize that a potential side-effect of robustness is that adversarial perturbations encourage better exploration of the environment.

## 5 DISCUSSION AND FUTURE WORK

Explicit distributional robustness of the form (5) is intractable except in limited cases. We provide a principled method for efficiently guaranteeing distributional robustness with a simple form of adversarial data perturbation. Using only assumptions about the smoothness of the loss function $\ell$, we prove that our method enjoys strong statistical guarantees and fast optimization rates for a large class of problems. The NP-hardness of certifying robustness for ReLU networks, coupled with our empirical success and theoretical certificates for smooth networks in deep learning, suggest that using smooth networks may be preferable if we wish to guarantee robustness. Empirical evaluations indicate that our methods are in fact robust to perturbations in the data, and they match or outperform less-principled adversarial training techniques. The major benefit of our approach is its simplicity and wide applicability across many models and machine-learning scenarios.

There remain many avenues for future investigation. Our optimization result (Theorem 2) applies only for small values of robustness $\rho$ and to a limited class of Wasserstein costs. Furthermore, our statistical guarantees (Theorems 3 and 4) use $\|\cdot\|_\infty$-covering numbers as a measure of model complexity, which can become prohibitively large for deep networks. In a learning-theoretic context, where the goal is to provide insight into convergence behavior as well as comfort that a procedure will "work" given enough data, such guarantees are satisfactory, but this may not be enough in security-essential contexts. This problem currently persists for most learning-theoretic guarantees in deep learning, and the recent works of Bartlett et al. (2017), Dziugaite & Roy (2017), and Neyshabur et al. (2017) attempt to mitigate this shortcoming. Replacing our current covering number arguments with more intricate notions such as margin-based bounds (Bartlett et al., 2017) would extend the scope and usefulness of our theoretical guarantees. Of course, the certificate (15) still holds regardless.

More broadly, this work focuses on small-perturbation attacks, and our theoretical guarantees show that it is possible to efficiently build models that provably guard against such attacks. Our method becomes another heuristic for protection against attacks with large adversarial budgets. Indeed, in the large-perturbation regime, efficiently training certifiably secure systems remains an important open question. We believe that conventional $\|\cdot\|_\infty$-defense heuristics developed for image classification do not offer much comfort in the large-perturbation/perceptible-attack setting: $\|\cdot\|_\infty$-attacks with a large budget can render images indiscernible to human eyes, while, for example, $\|\cdot\|_1$-attacks allow a concerted perturbation to critical regions of the image. Certainly $\|\cdot\|_\infty$-attack and defense models have been fruitful in building a foundation for security research in deep learning, but moving beyond them may be necessary for more advances in the large-perturbation regime.

ACKNOWLEDGMENTS

We thank Jacob Steinhardt for valuable feedback. AS, HN, and JD were partially supported by the SAIL-Toyota Center for AI Research. AS was also partially supported by a Stanford Graduate Fellowship and a Fannie & John Hertz Foundation Fellowship. HN was partially supported by a Samsung Fellowship. JD was partially supported by the National Science Foundation award NSF-CAREER-1553086.

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

## A ADDITIONAL EXPERIMENTS

### A.1 MNIST ATTACKS

We repeat Figure 3 using FGM (tow row of Figure 6) and IFGM (bottom row of Figure 6) attacks. The same trends are evident as in Figure 3.

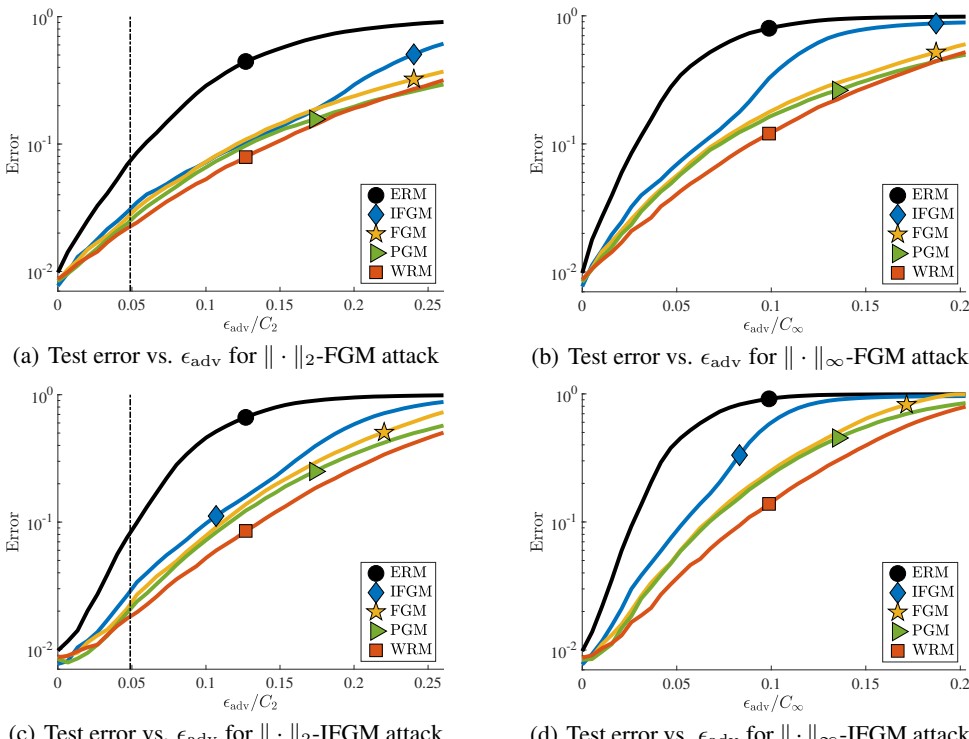

(a) Test error vs. $\epsilon_{\mathrm{adv}}$ for $\|\cdot\|_2$-FGM attack

(b) Test error vs. $\epsilon_{\mathrm{adv}}$ for $\|\cdot\|_\infty$-FGM attack

(c) Test error vs. $\epsilon_{\mathrm{adv}}$ for $\|\cdot\|_2$-IFGM attack

(d) Test error vs. $\epsilon_{\mathrm{adv}}$ for $\|\cdot\|_\infty$-IFGM attack

**Figure 6.** Further attacks on the MNIST dataset. We illustrate test misclassification error vs. the adversarial perturbation level $\epsilon_{\mathrm{adv}}$. Top row: FGM attacks, bottom row: IFGM attacks. Left column: Euclidean-norm attacks, right column: $\infty$-norm attacks. The vertical bar in (a) and (c) indicates the perturbation level that was used for training the PGM, FGM, and IFGM models and the estimated radius $\sqrt{\widehat{\rho}_n(\theta_{\mathrm{WRM}})}$.

### A.2 MNIST STABILITY OF LOSS SURFACE

In Figure 7, we repeat the illustration in Figure 4(b) for more digits. WRM's "misclassifications" are consistently reasonable to the human eye, as gradient-based perturbations actually transform the original image to other labels. Other models do not exhibit this behavior with the same consistency (if at all). Reasonable misclassifications correspond to having learned a data representation that makes gradients interpretable.

### A.3 MNIST EXPERIMENTS WITH VARIED $\gamma$

In Figure 8, we choose a fixed WRM adversary (fixed $\gamma_{\mathrm{adv}}$) and perturb WRM models trained with various penalty parameters $\gamma$. As the bound (11) with $\eta = \gamma$ suggests, even when the adversary has more budget than that used for training ($1/\gamma < 1/\gamma_{\mathrm{adv}}$), degradation in performance is still *smooth*. Further, as we decrease the penalty $\gamma$, the amount of achieved robustness—measured here by test error on adversarial perturbations with $\gamma_{\mathrm{adv}}$—has diminishing gains; this is again consistent with our theory which says that the inner problem (2b) is not efficiently computable for small $\gamma$.

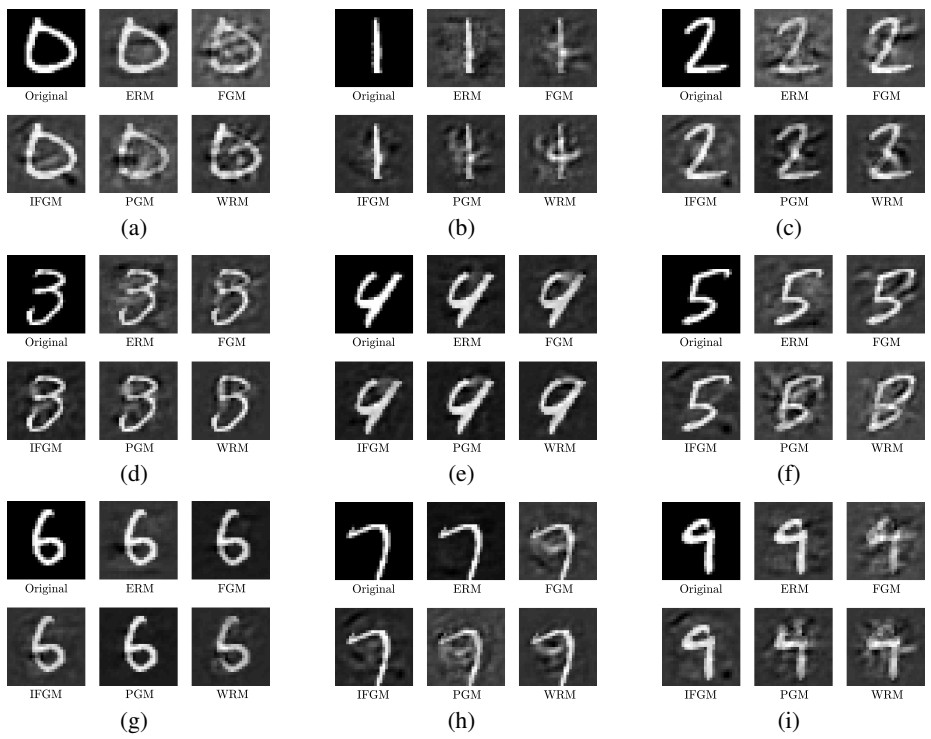

**Figure 7.** Visualizing stability over inputs. We illustrate the smallest WRM perturbation (largest $\gamma_{adv}$) necessary to make a model misclassify a datapoint.

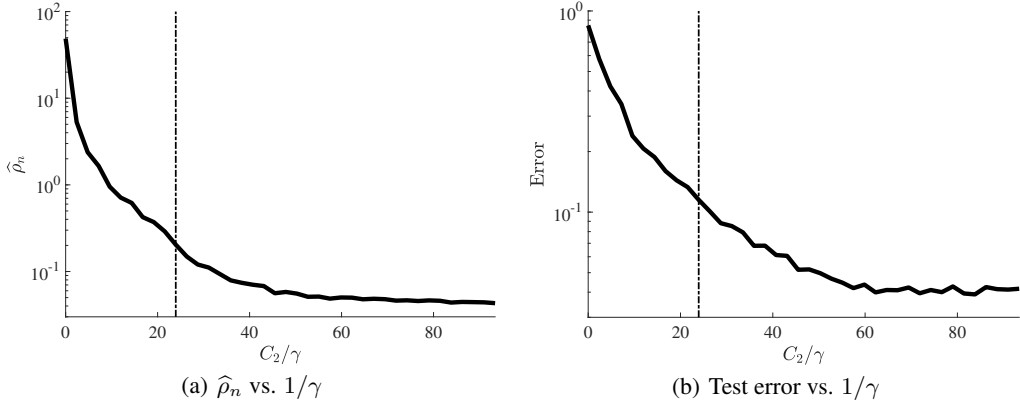

**Figure 8.** (a) Stability and (b) test error for a fixed adversary. We train WRM models with various levels of $\gamma$ and perturb them with a fixed WRM adversary ($\gamma_{adv}$ indicated by the vertical bar).

## A.4   MNIST EXPERIMENTS WITH A LARGER ADVERSARIAL BUDGET

Figures 9 and 10 repeat Figures 2(b), 3, and 6 for a larger training adversarial budget ($\gamma = 0.02C_2$) as well as larger test adversarial budgets. The distinctions in performance between various methods are less apparent now. For our method, the inner supremum is no longer strongly concave for over 10% of the data, indicating that we no longer have guarantees of performance. For large adversaries (i.e. large desired robustness values) our approach becomes a heuristic just like the other approaches.

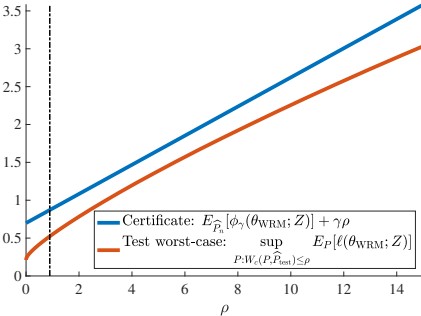

**Figure 9.** Empirical comparison between certificate of robustness (11) (blue) and out-of-sample (test) worst-case performance (red) for the experiments on MNIST with a larger training adversary. The statistical error term $\epsilon_n(t)$ is omitted from the certificate. The vertical bar indicates the achieved level of robustness on the training set $\widehat{\rho}_n(\theta_{\mathrm{WRM}})$.

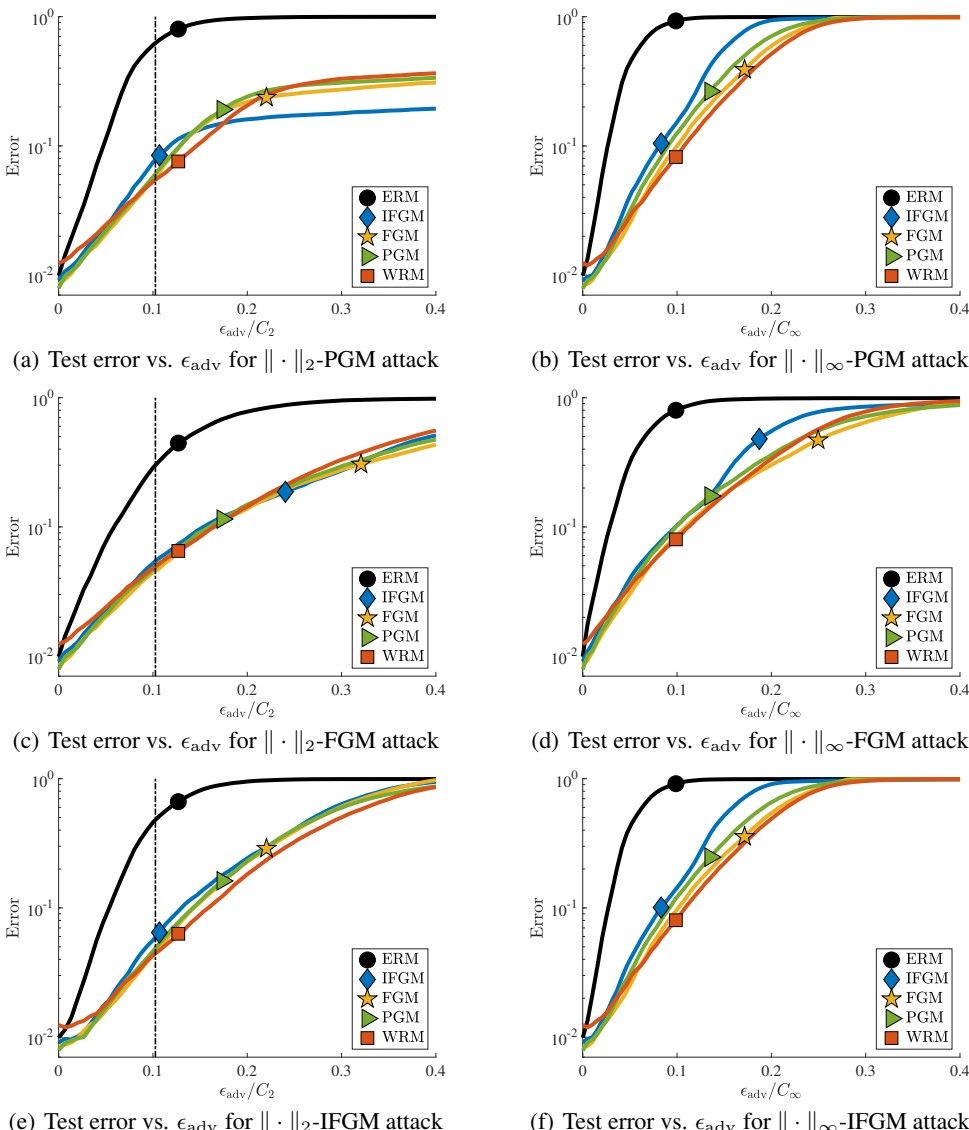

**Figure 10.** Attacks on the MNIST dataset with larger (training and test) adversarial budgets. We illustrate test misclassification error vs. the adversarial perturbation level $\epsilon_{\mathrm{adv}}$. Top row: PGM attacks, middle row: FGM attacks, bottom row: IFGM attacks. Left column: Euclidean-norm attacks, right column: $\infty$-norm attacks. The vertical bar in (a), (c), and (e) indicates the perturbation level that was used for training the PGM, FGM, and IFGM models and the estimated radius $\sqrt{\widehat{\rho}_n(\theta_{\mathrm{WRM}})}$.

## A.5 MNIST $\infty$-NORM EXPERIMENTS

We consider training FGM, IFGM, and PGM with $p = \infty$. We first compare with WRM trained in the same manner as before—with the squared Euclidean cost. Then, we consider a heuristic Lagrangian approach for training WRM with the squared $\infty$-norm cost.

### A.5.1 COMPARISON WITH STANDARD WRM

Our method (WRM) is trained to defend against $\|\cdot\|_2$-norm attacks by using the cost function

$$c((x,y),(x_0,y_0)) = \|x - x_0\|_2^2 + \infty \cdot \mathbf{1}\{y \neq y_0\}$$

with the convention that $0 \cdot \infty = 0$. Standard adversarial training methods often train to defend against $\|\cdot\|_\infty$-norm attacks, which we compare our method against in this subsection. Direct comparison between these approaches is not immediate, as we need to determine a suitable $\epsilon$ to train

FGM, IFGM, and PGM in the $\infty$-norm that corresponds to the penalty parameter $\gamma$ for the $\|\cdot\|_2$-norm that we use. Similar to the expression (19), we use

$$\epsilon := \mathbb{E}_{\widehat{P}_n}[\|T(\theta_{\mathrm{WRM}}, Z) - Z\|_\infty] \tag{23}$$

as the adversarial training budget for FGM, IFGM and PGM with $\|\cdot\|_\infty$-norms. Because 2-norm adversaries tend to focus budgets on a subset of features, the resulting $\infty$-norm perturbations are relatively large. In Figure 11 we show the results trained with a small training adversarial budget. In this regime, (large $\gamma$, small $\epsilon$), WRM matches the performance of other techniques.

In Figure 12 we show the results trained with a large training adversarial budget. In this regime (small $\gamma$, large $\epsilon$), performance between WRM and other methods diverge. WRM, which provably defends against small perturbations, outperforms other heuristics against imperceptible attacks for both Euclidean and $\infty$ norms. Further, it outperforms other heuristics on natural images, showing that it consistently achieves a smaller price of robustness. On attacks with large adversarial budgets (large $\epsilon_{\mathrm{adv}}$), however, the performance of WRM is worse than that of the other methods (especially in the case of $\infty$-norm attacks). These findings verify that WRM is a practical alternative over existing heuristics for the moderate levels of robustness where our guarantees hold.

### A.5.2 COMPARISON WITH $\|\cdot\|_\infty$-WRM

Our computational guarantees given in Theorem 2 does not hold anymore when we consider $\infty$-norm adversaries:

$$c((x, y), (x_0, y_0)) = \|x - x_0\|_\infty^2 + \infty \cdot \mathbf{1}\{y \neq y_0\}. \tag{24}$$

Optimizing the Lagrangian formulation (2b) with the $\infty$-norm is difficult since subtracting a multiple of the $\infty$-norm does not add (negative) curvature in all directions. In Appendix E, we propose a heuristic algorithm for solving the inner supremum problem (2b) with the above cost function (24). Our approach is based on a variant of proximal algorithms.

We compare our proximal heuristic introduced in Appendix E with other adversarial training procedures that were trained against $\infty$-norm adversaries. Results are shown in Figure 13 for a small training adversary and Figure 14 for a large training adversary. We observe that similar trends as in Section A.5.1 hold again.

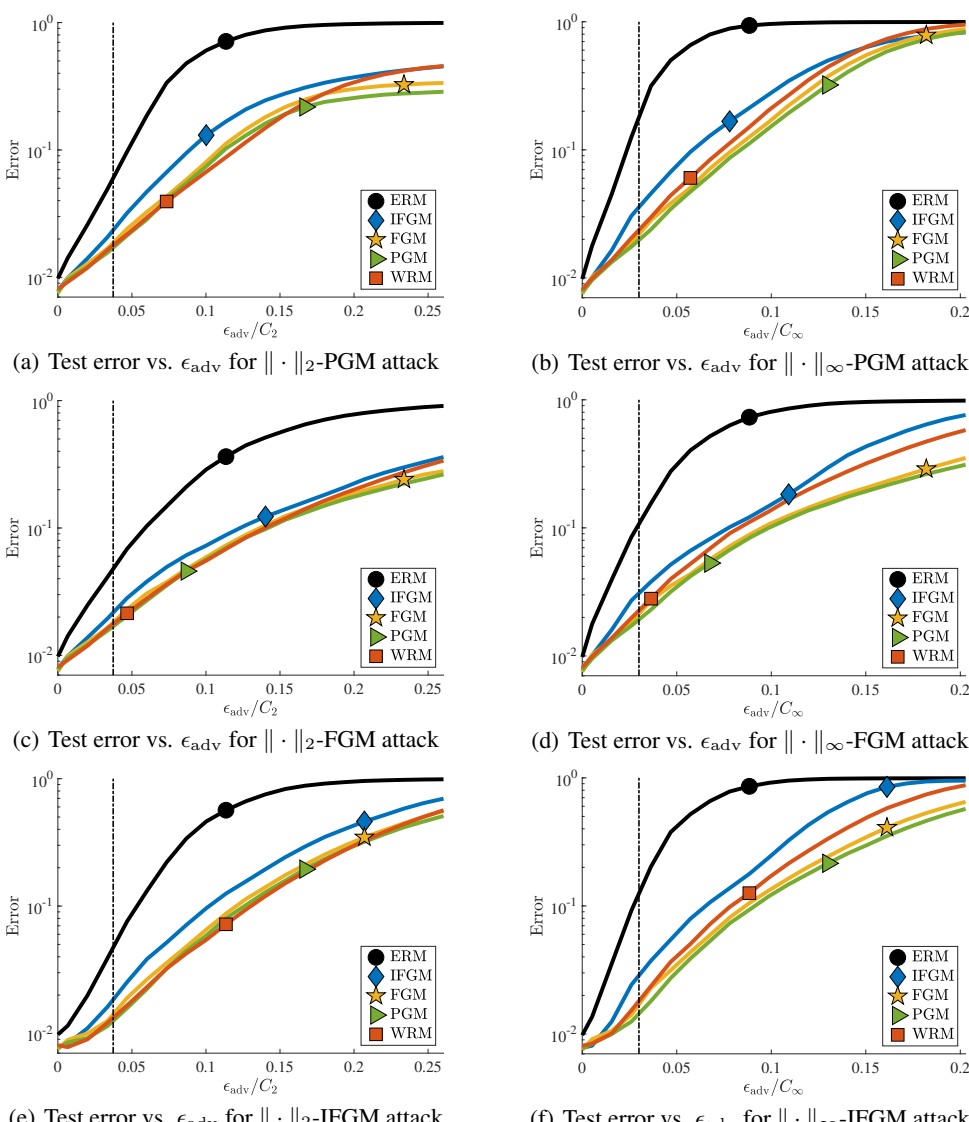

(a) Test error vs. $\epsilon_{\mathrm{adv}}$ for $\|\cdot\|_2$-PGM attack

(b) Test error vs. $\epsilon_{\mathrm{adv}}$ for $\|\cdot\|_\infty$-PGM attack

(c) Test error vs. $\epsilon_{\mathrm{adv}}$ for $\|\cdot\|_2$-FGM attack

(d) Test error vs. $\epsilon_{\mathrm{adv}}$ for $\|\cdot\|_\infty$-FGM attack

(e) Test error vs. $\epsilon_{\mathrm{adv}}$ for $\|\cdot\|_2$-IFGM attack

(f) Test error vs. $\epsilon_{\mathrm{adv}}$ for $\|\cdot\|_\infty$-IFGM attack

**Figure 11.** Attacks on the MNIST dataset. We compare standard WRM with $\infty$-norm PGM, FGM, IFGM. We illustrate test misclassification error vs. the adversarial perturbation level $\epsilon_{\mathrm{adv}}$. Top row: PGM attacks, middle row: FGM attacks, bottom row: IFGM attacks. Left column: Euclidean-norm attacks, right column: $\infty$-norm attacks. The vertical bar in (a), (c), and (e) indicates the estimated radius $\sqrt{\widehat{\rho}_n(\theta_{\mathrm{WRM}})}$. The vertical bar in (b), (d), and (f) indicates the perturbation level that was used for training the PGM, FGM, and IFGM models via (23).

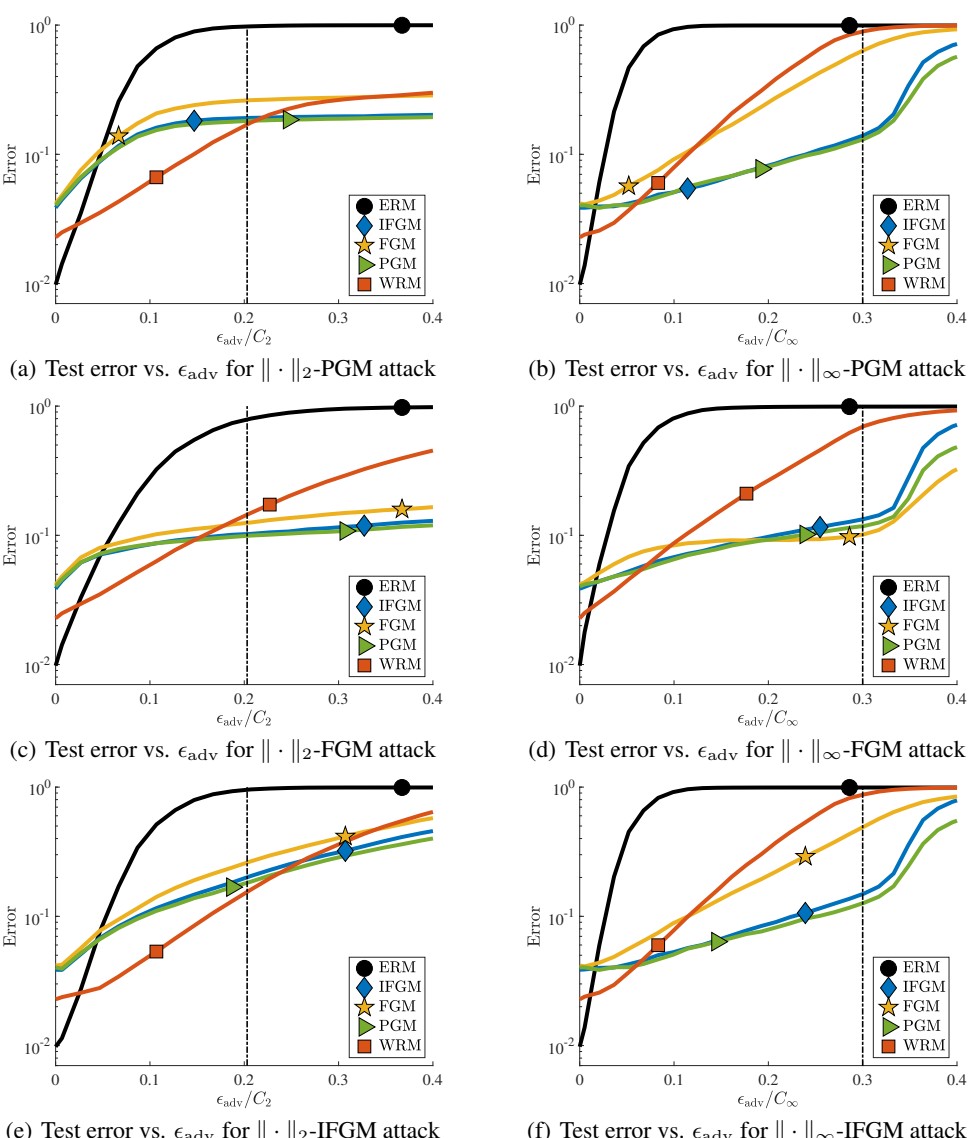

**Figure 12.** Attacks on the MNIST dataset with larger (training and test) adversarial budgets. We compare standard WRM with $\infty$-norm PGM, FGM, IFGM models. We illustrate test misclassification error vs. the adversarial perturbation level $\epsilon_{\mathrm{adv}}$. Top row: PGM attacks, middle row: FGM attacks, bottom row: IFGM attacks. Left column: Euclidean-norm attacks, right column: $\infty$-norm attacks. The vertical bar in (a), (c), and (e) indicates the estimated radius $\sqrt{\widehat{\rho}_n(\theta_{\mathrm{WRM}})}$. The vertical bar in (b), (d), and (f) indicates the perturbation level that was used for training the PGM, FGM, and IFGM models via (23).

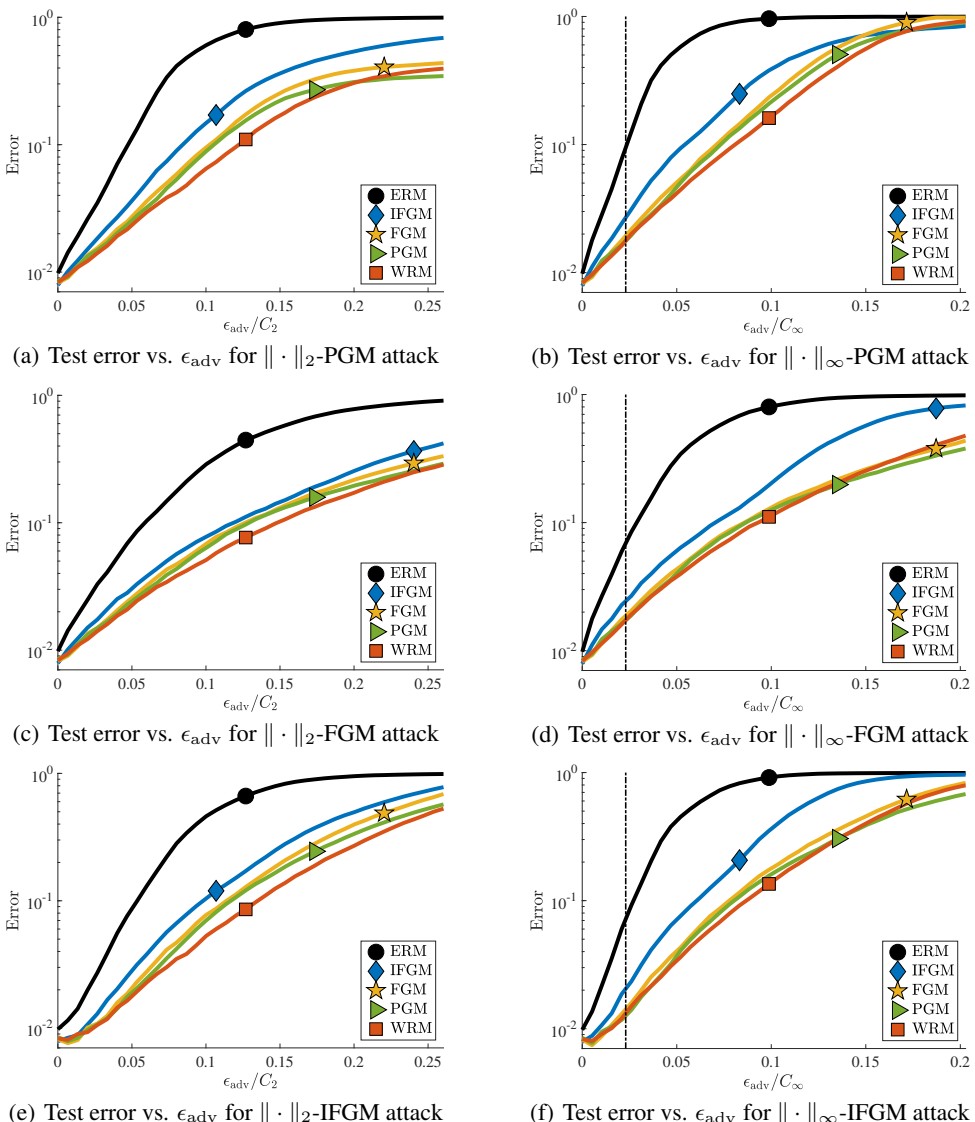

**Figure 13.** Attacks on the MNIST dataset. All models are trained in the $\infty$-norm. We illustrate test misclassification error vs. the adversarial perturbation level $\epsilon_{\mathrm{adv}}$. Top row: PGM attacks, middle row: FGM attacks, bottom row: IFGM attacks. Left column: Euclidean-norm attacks, right column: $\infty$-norm attacks. The vertical bar in (b), (d), and (f) indicates the perturbation level that was used for training the PGM, FGM, and IFGM models and the estimated radius $\sqrt{\widehat{\rho}_n(\theta_{\mathrm{WRM}})}$.

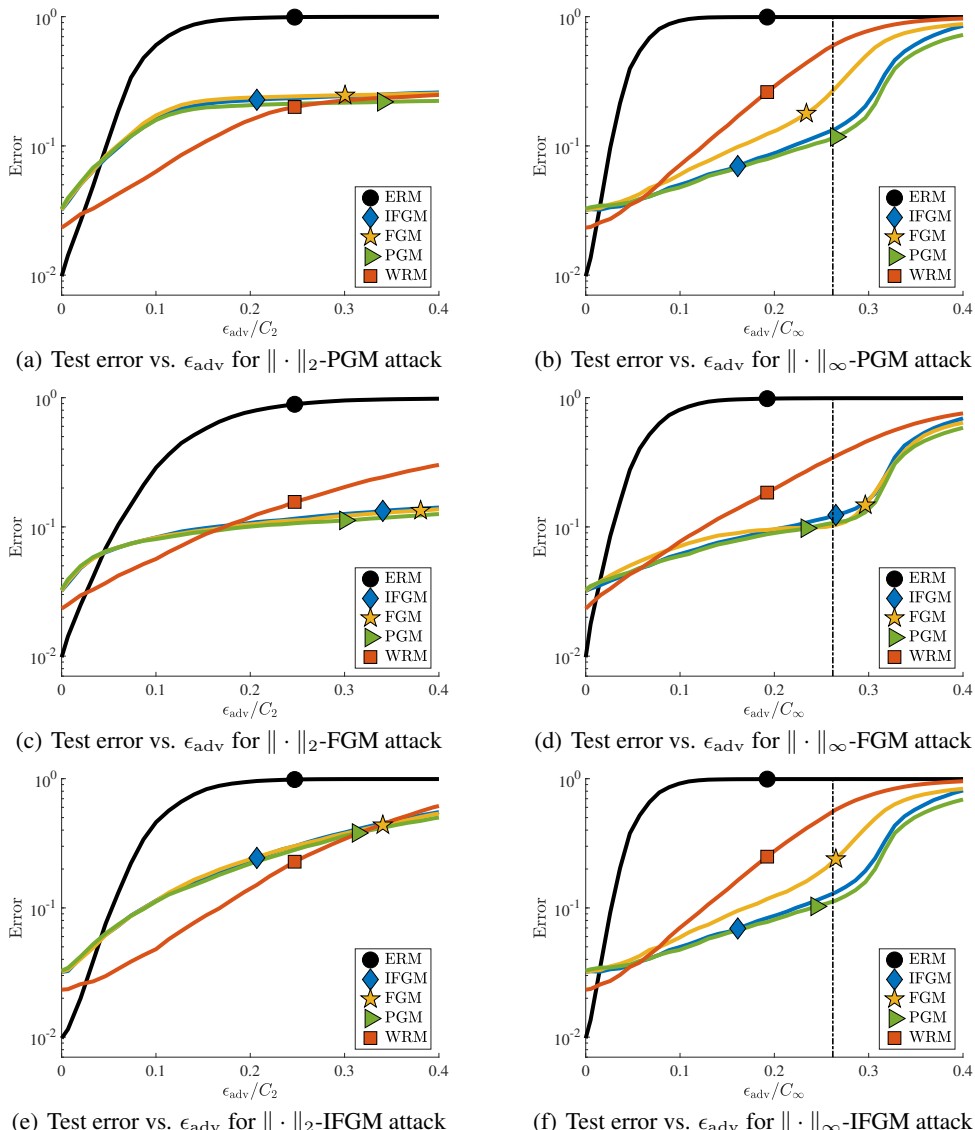

**Figure 14.** Attacks on the MNIST dataset with larger (training and test) adversarial budgets. All models are trained in the ∞-norm. We illustrate test misclassification error vs. the adversarial perturbation level $\epsilon_{\mathrm{adv}}$. Top row: PGM attacks, middle row: FGM attacks, bottom row: IFGM attacks. Left column: Euclidean-norm attacks, right column: ∞-norm attacks. The vertical bar in (b), (d), and (f) indicates the perturbation level that was used for training the PGM, FGM, and IFGM models and the estimated radius $\sqrt{\widehat{\rho}_n(\theta_{\mathrm{WRM}})}$.

## B    Finding worst-case perturbations with ReLU's is NP-hard

We show that computing worst-case perturbations $\sup_{u \in \mathcal{U}} \ell(\theta; z + u)$ is NP-hard for a large class of feedforward neural networks with ReLU activations. This result is essentially due to Katz et al. (2017a). In the following, we use polynomial time to mean polynomial growth with respect to $m$, the dimension of the inputs $z$.

An optimization problem is *NPO* (NP-Optimization) if (i) the dimensionality of the solution grows polynomially, (ii) the language $\{u \in \mathcal{U}\}$ can be recognized in polynomial time (i.e. a deterministic algorithm can decide in polynomial time whether $u \in \mathcal{U}$), and (iii) $\ell$ can be evaluated in polynomial time. We restrict analysis to feedforward neural networks with ReLU activations such that the cor-

responding worst-case perturbation problem is NPO.[4] Furthermore, we impose separable structure on $\mathcal{U}$, that is, $\mathcal{U} := \{v \le u \le w\}$ for some $v < w \in \mathbb{R}^m$.

**Lemma 2.** *Consider feedforward neural networks with ReLU's and let $\mathcal{U} := \{v \le u \le w\}$, where $v < w$ such that the optimization problem $\max_{u \in \mathcal{U}} \ell(\theta; z + u)$ is NPO. Then there exists $\theta$ such that this optimization problem is also NP-hard.*

**Proof**    First, we introduce the decision reformulation of the problem: for some $b$, we ask whether there exists some $u$ such that $\ell(\theta; z + u) \ge b$. The decision reformulation for an NPO problem is in NP, as a certificate for the decision problem can be verified in polynomial time. By appropriate scaling of $\theta$, $v$, and $w$, Katz et al. (2017a) show that 3-SAT Turing-reduces to this decision problem: given an oracle $D$ for the decision problem, we can solve an arbitrary instance of 3-SAT with a polynomial number of calls to $D$. The decision problem is thus NP-complete.

Now, consider an oracle $O$ for the optimization problem. The decision problem Turing-reduces to the optimization problem, as the decision problem can be solved with one call to $O$. Thus, the optimization problem is NP-hard. $\qquad\square$

## C    PROOFS

### C.1    PROOF OF PROPOSITION 1

For completeness, we provide an alternative proof to that given in Blanchet & Murthy (2016) using convex analysis. Our proof is less general, requiring the cost function $c$ to be continuous and convex in its first argument. The below general duality result gives Proposition 1 as an immediate special case. Recalling Rockafellar & Wets (1998, Def. 14.27 and Prop. 14.33), we say that a function $g : X \times Z \to \mathbb{R}$ is a *normal integrand* if for each $\alpha$, the mapping

$$z \mapsto \{x \mid g(x, z) \le \alpha\}$$

is closed-valued and measurable. We recall that if $g$ is continuous, then $g$ is a normal integrand (Rockafellar & Wets, 1998, Cor. 14.34); therefore, $g(x, z) = \gamma c(x, z) - \ell(\theta; x)$ is a normal integrand. We have the following theorem.

**Theorem 5.** *Let $f, c$ be such that for any $\gamma \ge 0$, the function $g(x, z) = \gamma c(x, z) - f(x)$ is a normal integrand. (For example, continuity of $f$ and closed convexity of $c$ is sufficient.) For any $\rho > 0$ we have*

$$\sup_{P:W_c(P,Q)} \int f(x) dP(x) = \inf_{\gamma \ge 0} \left\{ \int \sup_{x \in X} \{f(x) - \gamma c(x, z)\} \, dQ(z) + \gamma \rho \right\}.$$

**Proof**    First, the mapping $P \mapsto W_c(P, Q)$ is convex in the space of probability measures. As taking $P = Q$ yields $W_c(Q, Q) = 0$, Slater's condition holds and we may apply standard (infinite dimensional) duality results (Luenberger, 1969, Thm. 8.7.1) to obtain

$$\sup_{P:W_c(P,Q)} \int f(x) dP(x) = \sup_{P:W_c(P,Q)} \inf_{\gamma \ge 0} \left\{ \int f(x) dP(x) - \gamma W_c(P, Q) + \gamma \rho \right\}$$

$$= \inf_{\gamma \ge 0} \sup_{P:W_c(P,Q)} \left\{ \int f(x) dP(x) - \gamma W_c(P, Q) + \gamma \rho \right\}.$$

Now, noting that for any $M \in \Pi(P, Q)$ we have $\int f dP = \iint f(x) dM(x, z)$, we have that the rightmost quantity in the preceding display satisfies

$$\int f(x) dP(x) - \gamma \inf_{M \in \Pi(P,Q)} \int c(x, z) dM(x, z) = \sup_{M \in \Pi(P,Q)} \left\{ \int [f(x) - \gamma c(x, z)] dM(x, z) \right\}.$$

That is, we have

$$\sup_{P:W_c(P,Q)} \int f(x) dP(x) = \inf_{\gamma \ge 0} \sup_{P,M \in \Pi(P,Q)} \left\{ \int [f(x) - \gamma c(x, z)] dM(x, z) + \gamma \rho \right\}. \qquad (25)$$

---

[4]Note that $z, u \in \mathbb{R}^m$, so trivially the dimensionality of the solution grows polynomially.

Now, we note a few basic facts. First, because we have a joint supremum over $P$ and measures $M \in \Pi(P, Q)$ in expression (25), we have that

$$\sup_{P, M \in \Pi(P,Q)} \int [f(x) - \gamma c(x, z)] dM(x, z) \leq \int \sup_x [f(x) - \gamma c(x, z)] dQ(z).$$

We would like to show equality in the above. To that end, we note that if $\mathcal{P}$ denotes the space of regular conditional probabilities (Markov kernels) from $Z$ to $X$, then

$$\sup_{P, M \in \Pi(P,Q)} \int [f(x) - \gamma c(x, z)] dM(x, z) \geq \sup_{P \in \mathcal{P}} \int [f(x) - \gamma c(x, z)] dP(x \mid z) dQ(z).$$

Recall that a conditional distribution $P(\cdot \mid z)$ is regular if $P(\cdot \mid z)$ is a distribution for each $z$ and for each measurable $A$, the function $z \mapsto P(A \mid z)$ is measurable. Let $\mathcal{X}$ denote the space of all measurable mappings $z \mapsto x(z)$ from $Z$ to $X$. Using the powerful measurability results of Rockafellar & Wets (1998, Theorem 14.60), we have

$$\sup_{x \in \mathcal{X}} \int [f(x(z)) - \gamma c(x(z), z)] dQ(z) = \int \sup_{x \in X} [f(x) - \gamma c(x, z)] dQ(z)$$

because $f - c$ is upper semi-continuous, and the latter function is measurable. Now, let $x(z)$ be any measurable function that is $\epsilon$-close to attaining the supremum above. Define the conditional distribution $P(\cdot \mid z)$ to be supported on $x(z)$, which is evidently measurable. Then using the preceding display, we have

$$\int [f(x) - \gamma c(x, z)] dP(x \mid z) dQ(z) = \int [f(x(z)) - \gamma c(x(z), z)] dQ(z)$$

$$\geq \int \sup_{x \in X} [f(x) - \gamma c(x, z)] dQ(z) - \epsilon$$

$$\geq \sup_{P, M \in \Pi(P,Q)} \int [f(x) - \gamma c(x, z)] dM(x, z) - \epsilon.$$

As $\epsilon > 0$ is arbitrary, this gives

$$\sup_{P, M \in \Pi(P,Q)} \int [f(x) - \gamma c(x, z)] dM(x, z) = \int \sup_{x \in X} [f(x) - \gamma c(x, z)] dQ(z)$$

as desired, which implies both equality (6) and completes the proof. $\qquad \square$

## C.2   Proof of Lemma 1

First, note that $z^\star(\theta)$ is unique and well-defined by the strong convexity of $f(\theta, \cdot)$. For Lipschitzness of $z^\star(\theta)$, we first argue that $z^\star(\theta)$ is continuous in $\theta$. For any $\theta$, optimality of $z^\star(\theta)$ implies that $\mathsf{g_z}(\theta, z^\star(\theta))^T (z - z^\star(\theta)) \leq 0$. By strong concavity, for any $\theta_1, \theta_2$ and $z_1^\star = z^\star(\theta_1)$ and $z_2^\star = z^\star(\theta_2)$, we have

$$\frac{\lambda}{2} \|z_1^\star - z_2^\star\|^2 \leq f(\theta_2, z_2^\star) - f(\theta_2, z_1^\star) \text{ and } f(\theta_2, z_2^\star) \leq f(\theta_2, z_1^\star) + \mathsf{g_z}(\theta_2, z_1^\star)^T (z_2^\star - z_1^\star) - \frac{\lambda}{2} \|z_1^\star - z_2^\star\|^2.$$

Summing these inequalities gives

$$\lambda \|z_1^\star - z_2^\star\|^2 \leq \mathsf{g_z}(\theta_2, z_1^\star)^T (z_2^\star - z_1^\star) \leq (\mathsf{g_z}(\theta_2, z_1^\star) - \mathsf{g_z}(\theta_1, z_1^\star))^T (z_2^\star - z_1^\star),$$

where the last inequality follows because $\mathsf{g_z}(\theta_1, z_1^\star)^T (z_2^\star - z_1^\star) \leq 0$. Using a cross-Lipschitz condition from above and Holder's inequality, we obtain

$$\lambda \|z_1^\star - z_2^\star\|^2 \leq \|\mathsf{g_z}(\theta_2, z_1^\star) - \mathsf{g_z}(\theta_1, z_1^\star)\|_\star \|z_1^\star - z_2^\star\| \leq L_{\mathsf{z}\theta} \|\theta_1 - \theta_2\| \|z_1^\star - z_2^\star\|,$$

that is,

$$\|z_1^\star - z_2^\star\| \leq \frac{L_{\mathsf{z}\theta}}{\lambda} \|\theta_1 - \theta_2\|. \tag{26}$$

To see the second inequality, we show that $\bar{f}$ is differentiable with $\nabla \bar{f}(\theta) = g_\theta(\theta, z^\star(\theta))$. By using a variant of the envelope (or Danskin's) theorem, we first show directional differentiability of $\bar{f}$. Recall that we say $f$ is *inf-compact* if for all $\theta_0 \in \Theta$, there exists $\alpha > 0$ and a compact set $C \subset \Theta$ such that

$$\emptyset \neq \{z \in \mathcal{Z} : f(\theta, z) \leq \alpha\} \subset C$$

for all $\theta$ in some neighborhood of $\theta_0$ (Bonnans & Shapiro, 2013). See Bonnans & Shapiro (2013, Theorem 4.13) for a proof of the following result.

**Lemma 3.** *Suppose that $f(\cdot, z)$ is differentiable in $\theta$ for all $z \in \mathcal{Z}$, and $f$, $\nabla_z f$ are continuous on $\Theta \times \mathcal{Z}$. If $f$ is inf-compact, then $\bar{f}$ is directionally differentiable with*

$$\bar{f}'(\theta, d) = \sup_{z \in S(\theta)} \nabla_z f(\theta, z)^\top d$$

*where $S(\theta) = \operatorname{argmin}_z f(\theta, z)$.*

Now, note that from Assumption B, we have

$$|f(\theta, z) - f(\theta_0, z) - \nabla_\theta f(\theta_0, z)^\top (\theta - \theta_0)| \leq L_{\theta\theta} \|\theta - \theta_0\|$$

from which it is easy to see that $f$ is inf-compact. Applying Lemma 3 to $\bar{f}$ and noting that $S(\theta)$ is unique by strong convexity of $f(\theta, \cdot)$, we have that $\bar{f}$ is directionally differentiable with $\nabla \bar{f}(\theta) = g_\theta(\theta, z^\star(\theta))$. Since $g_\theta$ is continuous by Assumption B and $z^\star(\theta)$ is Lipschitz (26), we conclude that $\bar{f}$ is differentiable.

Finally, we have

$$\begin{aligned}
\|g_\theta(\theta_1, z_1^\star) - g_\theta(\theta_2, z_2^\star)\|_\star &\leq \|g_\theta(\theta_1, z_1^\star) - g_\theta(\theta_1, z_2^\star)\|_\star + \|g_\theta(\theta_1, z_2^\star) - g_\theta(\theta_2, z_2^\star)\|_\star \\
&\leq L_{\theta z} \|z_1^\star - z_2^\star\| + L_{\theta\theta} \|\theta_1 - \theta_2\| \\
&\leq \left( L_{\theta\theta} + \frac{L_{\theta z} L_{z\theta}}{\lambda} \right) \|\theta_1 - \theta_2\|,
\end{aligned}$$

where we have used inequality (26) again. This is the desired result.

### C.3  PROOF OF THEOREM 2

Our proof is based on that of Ghadimi & Lan (2013). For shorthand, let $f(\theta, z; z_0) = \ell(\theta; z) - \gamma c(z, z_0)$, noting that we perform gradient steps with

$$g^t = \nabla_\theta f(\theta^t, \hat{z}^t; z^t)$$

for $\hat{z}^t$ an $\epsilon$-approximate maximizer of $f(\theta, z; z^t)$ in $z$, and $\theta^{t+1} = \theta^t - \alpha_t g^t$. We assume $\alpha_t \leq \frac{1}{L_\phi}$ in the rest of the proof, which is satisfied for the constant stepsize $\alpha = \sqrt{\frac{\Delta_F}{L_\phi T \sigma^2}}$ and $T \geq \frac{L_\phi \Delta_F}{\sigma^2}$. By a Taylor expansion using the $L_\phi$-smoothness of the objective $F$, we have

$$\begin{aligned}
F(\theta^{t+1}) &\leq F(\theta^t) + \langle \nabla F(\theta^t), \theta^{t+1} - \theta^t \rangle + \frac{L_\phi}{2} \|\theta^{t+1} - \theta^t\|_2^2 \\
&= F(\theta^t) - \alpha_t \|\nabla F(\theta^t)\|_2^2 + \frac{L_\phi \alpha_t^2}{2} \|g^t\|_2^2 + \alpha_t \langle \nabla F(\theta^t), \nabla F(\theta^t) - g^t \rangle \\
&= F(\theta^t) - \alpha_t \left( 1 - \frac{1}{2} L_\phi \alpha_t \right) \|\nabla F(\theta^t)\|_2^2 \qquad (27) \\
&\quad + \alpha_t (1 - L_\phi \alpha_t) \langle \nabla F(\theta^t), \nabla F(\theta^t) - g^t \rangle + \frac{L_\phi \alpha_t^2}{2} \|g^t - \nabla F(\theta^t)\|_2^2.
\end{aligned}$$

Recalling the definition (2b) of $\phi_\gamma(\theta; z_0) = \sup_{z \in \mathcal{Z}} f(\theta, z; z_0)$, we define the potentially biased errors $\delta^t = g^t - \nabla_\theta \phi_\gamma(\theta^t; z^t)$. Substituting the this into the progress guarantee (27), we have

$$
F(\theta^{t+1}) \leq F(\theta^t) - \alpha_t \left(1 - \frac{1}{2} L_\phi \alpha_t\right) \left\|\nabla F(\theta^t)\right\|_2^2 + \alpha_t \left(1 - L_\phi \alpha_t\right) \left\langle \nabla F(\theta^t), \nabla F(\theta^t) - \nabla_\theta \phi_\gamma(\theta; z^t) \right\rangle
$$

$$
- \alpha_t \left(1 - L_\phi \alpha_t\right) \left\langle \nabla F(\theta^t), \delta^t \right\rangle + \frac{L_\phi \alpha_t^2}{2} \left\|\nabla_\theta \phi_\gamma(\theta; z^t) + \delta^t - \nabla F(\theta^t)\right\|_2^2
$$

$$
= F(\theta^t) - \alpha_t \left(1 - \frac{1}{2} L_\phi \alpha_t\right) \left\|\nabla F(\theta^t)\right\|_2^2 + \alpha_t \left(1 - L_\phi \alpha_t\right) \left\langle \nabla F(\theta^t), \nabla F(\theta^t) - \nabla_\theta \phi_\gamma(\theta; z^t) \right\rangle
$$

$$
- \alpha_t \left(1 - L_\phi \alpha_t\right) \left\langle \nabla F(\theta^t), \delta^t \right\rangle
$$

$$
+ \frac{L_\phi \alpha_t^2}{2} \left( \left\|\delta^t\right\|_2^2 + \left\|\nabla_\theta \phi_\gamma(\theta; z^t) - \nabla F(\theta^t)\right\|_2^2 + 2 \left\langle \nabla_\theta \phi_\gamma(\theta; z^t) - \nabla F(\theta^t), \delta^t \right\rangle \right).
$$

Using $\pm \langle a, b \rangle \leq \frac{1}{2} \|a\|_2^2 + \frac{1}{2} \|b\|_2^2$ in the preceding display, we get

$$
F(\theta^{t+1}) \leq F(\theta^t) - \frac{\alpha_t}{2} \left\|\nabla F(\theta^t)\right\|_2^2 + \alpha_t \left(1 - L_\phi \alpha_t\right) \left\langle \nabla F(\theta^t), \nabla F(\theta^t) - \nabla_\theta \phi_\gamma(\theta; z^t) \right\rangle
$$

$$
+ \frac{\alpha_t \left(1 + L_\phi \alpha_t\right)}{2} \left\|\delta^t\right\|_2^2 + L_\phi \alpha_t^2 \left\|\nabla_\theta \phi_\gamma(\theta; z^t) - \nabla F(\theta^t)\right\|_2^2 \tag{28}
$$

Letting $z_\star^t = \operatorname{argmax}_z f(\theta^t, z; z^t)$, note that the error $\delta^t$ satisfies

$$
\left\|\delta^t\right\|_2^2 = \left\|\nabla_\theta \phi_\gamma(\theta^t; z^t) - \nabla_\theta f(\theta, \widehat{z}^t; z^t)\right\|_2^2 = \left\|\nabla_\theta \ell(\theta, z_\star^t) - \nabla_\theta \ell(\theta, \widehat{z}^t)\right\|_2^2
$$

$$
\leq L_{\theta z}^2 \|\widehat{z}^t - z_\star^t\|_2^2 \leq \frac{2 L_{\theta z}^2}{\lambda} \epsilon,
$$

where the final inequality uses the $\lambda = \gamma - L_{zz}$ strong-concavity of $z \mapsto f(\theta, z; z_0)$. For shorthand, let $\widehat{\epsilon} = \frac{2 L_{\theta z}^2}{\gamma - L_{zz}} \epsilon$. Taking conditional expectations in the bound (28) and using $\mathbb{E}[\nabla_\theta \phi_\gamma(\theta^t; z^t) \mid \theta^t] = \nabla F(\theta^t)$, we have

$$
\mathbb{E}[F(\theta^{t+1}) - F(\theta^t) \mid \theta^t] \leq -\frac{\alpha_t}{2} \left\|\nabla F(\theta^t)\right\|_2^2 + \frac{\alpha_t(1 + L_\phi \alpha_t)}{2} \widehat{\epsilon} + L_\phi \alpha_t^2 \left\|\nabla_\theta \phi_\gamma(\theta; z^t) - \nabla F(\theta^t)\right\|_2^2
$$

$$
\leq -\frac{\alpha_t}{2} \left\|\nabla F(\theta^t)\right\|_2^2 + \alpha_t \widehat{\epsilon} + L_\phi \alpha_t^2 \left\|\nabla_\theta \phi_\gamma(\theta; z^t) - \nabla F(\theta^t)\right\|_2^2,
$$

where we use the fact that $\alpha_t \leq \frac{1}{L_\phi}$. For a fixed stepsize $\alpha$, taking the total expectation yields

$$
\mathbb{E}\left[\left\|\nabla F(\theta^t)\right\|_2^2\right] - 2\widehat{\epsilon} \leq \frac{2}{\alpha} \mathbb{E}[F(\theta^t) - F(\theta^{t+1})] + 2 L_\phi \alpha \sigma^2
$$

since we have $\mathbb{E}[\|\nabla \phi_\gamma(\theta; Z) - \nabla F(\theta)\|_2^2] \leq \sigma^2$ by assumption. Summing over $t$, we have

$$
\frac{1}{T} \sum_{t=0}^{T-1} \mathbb{E}\left[\left\|\nabla F(\theta^t)\right\|_2^2\right] - 2\widehat{\epsilon} \leq \frac{2}{\alpha T} \left(F(\theta^0) - \mathbb{E}[F(\theta^T)]\right) + 2 L_\phi \alpha \sigma^2
$$

$$
\leq \frac{2 \Delta_F}{\alpha T} + 2 L_\phi \alpha \sigma^2,
$$

where the latter inequality holds since $\inf_\theta F(\theta) \leq F(\theta^T)$. Plugging in $\alpha = \sqrt{\frac{\Delta_F}{L_\phi T \sigma^2}}$ gives the result.

## C.4 PROOF OF THEOREM 3

We first show the bound (11). From the duality result (5), we have the deterministic result that

$$
\sup_{P : W_c(P, Q) \leq \rho} \mathbb{E}_Q[\ell(\theta; Z)] \leq \gamma \rho + \mathbb{E}_Q[\phi_\gamma(\theta; Z)]
$$

for all $\rho > 0$, distributions $Q$, and $\gamma \geq 0$. Next, we show that $\mathbb{E}_{\widehat{P}_n}[\phi_\gamma(\theta; Z)]$ concentrates around its population counterpart at the usual rate (Boucheron et al., 2005).

First, we have that

$$\phi_\gamma(\theta; z) \in [-M_\ell, M_\ell],$$

because $-M_\ell \leq \ell(\theta; z) \leq \phi_\gamma(\theta; z) \leq \sup_z \ell(\theta; z) \leq M_\ell$. Thus, the functional $\theta \mapsto F_n(\theta)$ satisfies bounded differences (Boucheron et al., 2013, Thm. 6.2), and applying standard results on Rademacher complexity (Bartlett & Mendelson, 2002) and entropy integrals (van der Vaart & Wellner, 1996, Ch. 2.2) gives the result.

To see the second result (12), we substitute $\rho = \widehat{\rho}_n$ in the bound (11). Then, with probability at least $1 - e^{-t}$, we have

$$\sup_{P:W_c(P,P_0) \leq \widehat{\rho}_n(\theta)} \mathbb{E}_P[\ell(\theta; Z)] \leq \gamma \widehat{\rho}_n(\theta) + \mathbb{E}_{\widehat{P}_n}[\phi_\gamma(\theta; Z)] + \epsilon_n(t).$$

Since we have

$$\sup_{P:W_c(P,\widehat{P}_n) \leq \widehat{\rho}_n(\theta)} \mathbb{E}_P[\ell(\theta; Z)] = \mathbb{E}_{\widehat{P}_n}[\phi_\gamma(\theta; Z)] + \gamma \widehat{\rho}_n(\theta).$$

from the strong duality in Proposition 1, our second result follows.

### C.5 PROOF OF COROLLARY 1

The result is essentially standard (van der Vaart & Wellner, 1996), which we now give for completeness. Note that for $\mathcal{F} = \{\ell(\theta; \cdot) : \theta \in \Theta\}$, any $(\epsilon, \|\cdot\|)$-covering $\{\theta_1, \ldots, \theta_N\}$ of $\Theta$ guarantees that $\min_i |\ell(\theta; z) - \ell(\theta_i; z)| \leq L\epsilon$ for all $\theta, z$, or

$$N(\mathcal{F}, \epsilon, \|\cdot\|_{L^\infty(\mathcal{Z})}) \leq N(\Theta, \epsilon/L, \|\cdot\|) \leq \left(1 + \frac{\text{diam}(\Theta)L}{\epsilon}\right)^d,$$

where $\text{diam}(\Theta) = \sup_{\theta, \theta' \in \Theta} \|\theta - \theta'\|$. Noting that $|\ell(\theta; Z)| \leq L\,\text{diam}(\Theta) + M_0 =: M_\ell$, we have the result.

### C.6 PROOF OF THEOREM 4

Define

$$P_n^*(\theta) := \underset{P}{\text{argmax}} \left\{\mathbb{E}_P[\ell(\theta; Z)] - \gamma W_c(P, \widehat{P}_n)\right\},$$

$$P^*(\theta) := \underset{P}{\text{argmax}} \left\{\mathbb{E}_P[\ell(\theta; Z)] - \gamma W_c(P, P_0)\right\}.$$

First, we show that $P^*(\theta)$ and $P_n^*(\theta)$ are attained for all $\theta \in \Theta$. We omit the dependency on $\theta$ for notational simplicity and only show the result for $P^*(\theta)$ as the case for $P_n^*(\theta)$ is symmetric. Let $P^\epsilon$ be an $\epsilon$-maximizer, so that

$$\mathbb{E}_{P^\epsilon}[\ell(\theta; Z)] - \gamma W_c(P^\epsilon, P_0) \geq \sup_P \left\{\mathbb{E}_P[\ell(\theta; Z)] - \gamma W_c(P_n, P_0)\right\} - \epsilon.$$

As $\mathcal{Z}$ is compact, the collection $\{P^{1/k}\}_{k \in \mathbb{N}}$ is a uniformly tight collection of measures. By Prohorov's theorem (Billingsley, 1999, Ch 1.1, p. 57), (restricting to a subsequence if necessary), there exists some distribution $P^*$ on $\mathcal{Z}$ such that $P^{1/k} \overset{d}{\to} P^*$ as $k \to \infty$. Continuity properties of Wasserstein distances (Villani, 2009, Corollary 6.11) then imply that

$$\lim_{k \to \infty} W_c(P^{1/k}, P_0) = W_c(P^*, P_0). \tag{29}$$

Combining (29) and the monotone convergence theorem, we obtain

$$\mathbb{E}_{P^*}[\ell(\theta; Z)] - \gamma W_c(P^*, P_0) = \lim_{k \to \infty} \left\{\mathbb{E}_{P^{1/k}}[\ell(\theta; Z)] - \gamma W_c(P^{1/k}, P_0)\right\}$$
$$\geq \sup_P \left\{\mathbb{E}_P[\ell(\theta; Z)] - \gamma W_c(P, P_0)\right\}.$$

We conclude that $P^*$ is attained for all $P_0$.

Next, we show the concentration result (30). Recall the definition (9) of the transportation mapping

$$T(\theta, z) := \operatorname*{argmax}_{z' \in \mathcal{Z}} \{\ell(\theta; z') - \gamma c(z', z)\},$$

which is unique and well-defined under our strong concavity assumption that $\gamma > L_{zz}$, and smooth (recall Eq. (16)) in $\theta$. Then by Proposition 1 (or by using a variant of Kantorovich duality (Villani, 2009, Chs. 9–10)), we have

$$\mathbb{E}_{P_n^*(\theta)}[\ell(\theta; Z) = \mathbb{E}_{\widehat{P}_n}[\ell(\theta; T(\theta; Z))] \text{ and } \mathbb{E}_{P^*(\theta)}[\ell(\theta; Z) = \mathbb{E}_{P_0}[\ell(\theta; T(\theta; Z))]$$
$$W_c(P_n^*(\theta), \widehat{P}_n) = \mathbb{E}_{\widehat{P}_n}[c(T(\theta; Z), Z)] \text{ and } W_c(P^*(\theta), P_0) = \mathbb{E}_{P_0}[c(T(\theta; Z), Z)].$$

We now proceed by showing the uniform convergence of

$$\mathbb{E}_{\widehat{P}_n}[c(T(\theta; Z), Z)] \quad \text{to} \quad \mathbb{E}_{P_0}[c(T(\theta; Z), Z)]$$

under both cases (i), that $c$ is Lipschitz, and (ii), that $\ell$ is Lipschitz in $z$, using a covering argument on $\Theta$. Recall inequality (16) (i.e. Lemma 1), which is that

$$\|T(\theta_1; z) - T(\theta_2; z)\| \leq \frac{L_{z\theta}}{[\gamma - L_{zz}]_+} \|\theta_1 - \theta_2\|.$$

We have the following lemma.

**Lemma 4.** *Assume the conditions of Theorem 4. Then for any $\theta_1, \theta_2 \in \Theta$,*

$$|c(T(\theta_1; z), z) - c(T(\theta_2; z), z)| \leq \frac{L_c L_{z\theta}}{[\gamma - L_{zz}]_+} \|\theta_1 - \theta_2\|.$$

**Proof** In the first case, that $c$ is $L_c$-Lipschitz in its first argument, this is trivial: we have

$$|c(T(\theta_1; z), z) - c(T(\theta_2; z), z)| \leq L_c \|T(\theta_1; z) - T(\theta_2; z)\| \leq \frac{L_c L_{z\theta}}{[\gamma - L_{zz}]_+} \|\theta_1 - \theta_2\|$$

by the smoothness inequality (16) for $T$.

In the second case, that $z \mapsto \ell(\theta, z)$ is $L_c$-Lipschitz, let $z_i = T(\theta_i; z)$ for shorthand. Then we have

$$\gamma c(z_2, z) - \gamma c(z_1, z) = \gamma c(z_2, z) - \ell(\theta_2, z_2) + \ell(\theta_2, z_2) - \gamma c(z_1, z)$$
$$\leq \gamma c(z_1, z) - \ell(\theta_2, z_1) + \ell(\theta_2, z_2) - \gamma c(z_1, z) = \ell(\theta_2, z_2) - \ell(\theta_2, z_1),$$

and similarly,

$$\gamma c(z_2, z) - \gamma c(z_1, z) = \gamma c(z_2, z) - \ell(\theta_1, z_1) + \ell(\theta_1, z_1) - \gamma c(z_1, z)$$
$$\geq \gamma c(z_2, z) - \ell(\theta_1, z_1) + \ell(\theta_1, z_2) - \gamma c(z_2, z) = \ell(\theta_1, z_2) - \ell(\theta_1, z_1).$$

Combining these two inequalities and using that

$$|\ell(\theta, z_2) - \ell(\theta, z_1)| \leq \gamma L_c \|z_2 - z_1\|$$

for any $\theta$ gives the result. $\qquad\square$

Using Lemma 4 we obtain that $\theta \mapsto |\mathbb{E}_{\widehat{P}_n}[c(T(\theta; Z), \theta)] - \mathbb{E}_{P_0}[c(T(\theta; Z), Z)]|$ is $2L_c L_{z\theta}/[\gamma - L_{zz}]_+$-Lipschitz. Let $\Theta_{\text{cover}} = \{\theta_1, \cdots, \theta_N\}$ be a $\frac{[\gamma - L_{zz}]_+ t}{4 L_c L_{z\theta}}$-cover of $\Theta$ with respect to $\|\cdot\|$. From Lipschitzness of $|\mathbb{E}_{\widehat{P}_n}[c(T(\theta; Z), Z)] - \mathbb{E}_{P_0}[c(T(\theta; Z), Z)]|$, we have that if for all $\theta \in \{\Theta_{\text{cover}}\}$,

$$|\mathbb{E}_{\widehat{P}_n}[c(T(\theta; Z), Z)] - \mathbb{E}_{P_0}[c(T(\theta; Z), \theta)]| \leq \frac{t}{2},$$

then it follows that

$$\sup_{\theta \in \Theta} |\mathbb{E}_{\widehat{P}_n}[c(T(\theta; Z), Z)] - \mathbb{E}_{P_0}[c(T(\theta; Z), Z)]| \leq t.$$

Under the first assumption $(i)$, we have $|c(T(\theta; Z), Z)| \leq 2L_{\mathsf{c}}M_{\mathsf{z}}$. Applying Hoeffding's inequality, for any fixed $\theta \in \Theta$

$$\mathbb{P}\left(|\mathbb{E}_{\widehat{P}_n}[c(T(\theta; Z), Z)] - \mathbb{E}_{P_0}[c(T(\theta; Z), Z)]| \geq \frac{t}{2}\right) \leq 2\exp\left(-\frac{nt^2}{32L_{\mathsf{c}}^2 M_{\mathsf{z}}^2}\right).$$

Taking a union bound over $\theta_1, \cdots, \theta_N$, we conclude that

$$\mathbb{P}\left(\sup_{\theta \in \Theta}|\mathbb{E}_{\widehat{P}_n}[c(T(\theta; Z), Z)] - \mathbb{E}_{P_0}[c(T(\theta; Z), Z)]| \geq t\right) \leq 2N\left(\Theta, \frac{[\gamma - L_{\mathsf{zz}}]_+ \, t}{4L_{\mathsf{c}}L_{\mathsf{z}\theta}}, \|\cdot\|\right)\exp\left(-\frac{nt^2}{32L_{\mathsf{c}}^2 M_{\mathsf{z}}^2}\right)$$

which was our desired result (30).

Under the second assumption $(ii)$, we have from the definition of the transport map $T$

$$\gamma c(T(\theta; z), z) \leq \ell(\theta; z) \leq M_\ell$$

and hence $|c(T(\theta; Z), Z)| \leq M_\ell/\gamma$. The result for the second case follows from an identical reasoning.

# D  SUPERVISED LEARNING

In supervised learning settings, it is often natural—for example, in classification—to only consider adversarial perturbations to the feature vectors (covariates). In this section, we give an adapation of the results in Sections 2 and 3 (Theorems 2 and 4) to such scenarios. Let $Z = (X, Y) \in \mathcal{X} \times \mathbb{R}$ where $X \in \mathcal{X}$ is a feature vector[5] and $Y \in \mathbb{R}$ is a label. In classification settings, we have $Y \in \{1, \ldots, K\}$. We consider an adversary that can only perturb the feature vector $X$ (Goodfellow et al., 2015), which can be easily represented in our robust formulation (2) by defining the cost function $c : \mathcal{Z} \times \mathcal{Z} \rightarrow \mathbb{R}_+ \cup \{\infty\}$ as follows: for $z = (x, y)$ and $z' = (x', y')$, recall the covariate shift cost function (8)

$$c(z, z') := c_x(x, x') + \infty \cdot \mathbf{1}\{y \neq y'\},$$

where $c_x : \mathcal{X} \times \mathcal{X} \rightarrow \mathbb{R}_+$ is the transportation cost for the feature vector $X$. As before, we assume that $c_x$ is nonnegative, continuous, convex in its first argument and satisfies $c_x(x, x) = 0$.

Under the cost function (8), the robust surrogate loss in the penalty problem (2) and its empirical counterpart (7) becomes

$$\phi_\gamma(\theta; (x_0, y_0)) = \sup_{x \in \mathcal{X}}\{\ell(\theta; (x, y_0)) - \gamma c_x(x, x_0)\}.$$

Similarly as in Section 2.1, we require the following two assumptions that guarantee efficient computability of the robust surrogate $\phi_\gamma$.

**Assumption C.** *The function $c_x : \mathcal{X} \times \mathcal{X} \rightarrow \mathbb{R}_+$ is continuous. For each $x_0 \in \mathcal{X}$, $c_x(\cdot, x_0)$ is 1-strongly convex with respect to the norm $\|\cdot\|$.*

Let $\|\cdot\|_*$ be the dual norm to $\|\cdot\|$; we again abuse notation by using the same norm $\|\cdot\|$ on $\Theta$ and $\mathcal{X}$, though the specific norm is clear from context.

**Assumption D.** *The loss $\ell : \Theta \times \mathcal{Z} \rightarrow \mathbb{R}$ satisfies the Lipschitzian smoothness conditions*

$$\|\nabla_\theta \ell(\theta; (x, y)) - \nabla_\theta \ell(\theta'; (x, y))\|_* \leq L_{\theta\theta}\|\theta - \theta'\|, \quad \|\nabla_x \ell(\theta; (x, y)) - \nabla_x \ell(\theta; (x', y))\|_* \leq L_{\mathsf{xx}}\|x - x'\|,$$

$$\|\nabla_\theta \ell(\theta; (x, y)) - \nabla_\theta \ell(\theta; (x', y))\|_* \leq L_{\theta\mathsf{x}}\|x - x'\|, \quad \|\nabla_x \ell(\theta; (x, y)) - \nabla_x \ell(\theta'; (x, y))\|_* \leq L_{\mathsf{x}\theta}\|\theta - \theta'\|.$$

Under Assumptions C and D, an analogue of Lemma 1 still holds. The proof of the following result is nearly identical to that of Lemma 1; we state the full result for completeness.

**Lemma 5.** *Let $f : \Theta \times \mathcal{X} \rightarrow \mathbb{R}$ be differentiable and $\lambda$-strongly concave in $x$ with respect to the norm $\|\cdot\|$, and define $\bar{f}(\theta) = \sup_{x \in \mathcal{X}} f(\theta, x)$. Let $\mathsf{g}_\theta(\theta, x) = \nabla_\theta f(\theta, x)$ and $\mathsf{g}_\mathsf{x}(\theta, x) = \nabla_x f(\theta, x)$, and assume $\mathsf{g}_\theta$ and $\mathsf{g}_\mathsf{x}$ satisfy the Lipschitz conditions of Assumption B. Then $\bar{f}$ is differentiable, and letting $x^\star(\theta) = \mathrm{argmax}_{x \in \mathcal{X}} f(\theta, x)$, we have $\nabla \bar{f}(\theta) = \mathsf{g}_\theta(\theta, x^\star(\theta))$. Moreover,*

$$\|x^\star(\theta_1) - x^\star(\theta_2)\| \leq \frac{L_{\mathsf{x}\theta}}{\lambda}\|\theta_1 - \theta_2\| \quad \text{and} \quad \|\nabla\bar{f}(\theta) - \nabla\bar{f}(\theta')\|_\star \leq \left(L_{\theta\theta} + \frac{L_{\theta\mathsf{x}}L_{\mathsf{x}\theta}}{\lambda}\right)\|\theta - \theta'\|.$$

---

[5]We assume that $\mathcal{X}$ is a subset of normed vector space.

From Lemma 5, our previous results (Theorems 2 and 4) follow. The following is an analogue of Theorem 2 for the cost function (8).

**Theorem 6** (Convergence of Nonconvex SGD). *Let Assumptions C and D hold with the $\ell_2$-norm and let $\Theta = \mathbb{R}^d$. Let $\Delta_F \geq F(\theta^0) - \inf_\theta F(\theta)$. Assume $\mathbb{E}[\|\nabla F(\theta) - \nabla_\theta \phi_\gamma(\theta, Z)\|_2^2] \leq \sigma^2$ and take constant stepsizes $\alpha = \sqrt{\frac{\Delta_F}{L_\phi T \sigma^2}}$ where $L_\phi := L_{\theta\theta} + \frac{L_{\theta x} L_{x\theta}}{\gamma - L_{xx}}$. For $T \geq \frac{L_\phi \Delta_F}{\sigma^2}$, Algorithm 1 satisfies*

$$\frac{1}{T} \sum_{t=0}^{T-1} \mathbb{E}\left[\|\nabla F(\theta^t)\|_2^2\right] - \frac{4L_{\theta x}^2}{\gamma - L_{xx}}\epsilon \leq 4\sigma\sqrt{\frac{L_\phi \Delta_F}{T}}.$$

Similarly, an analogous result to Theorem 4 holds. Define the transport map for the covariate shift

$$T_\gamma(\theta; (x_0, y_0)) := \operatorname*{argmax}_{x \in \mathcal{X}}\{\ell(\theta; (x, y_0)) - \gamma c_x(x, x_0)\}.$$

**Theorem 7.** *Let $\mathcal{Z} \subset \{z \in \mathbb{R}^m : \|z\| \leq M_z\}$ so that $\|Z\| \leq M_z$ almost surely and assume either that (i) $c_x(\cdot, \cdot)$ is $L_c$-Lipschitz over $\mathcal{X}$ with respect to the norm $\|\cdot\|$ in each argument, or (ii) that $\ell(\theta, z) \in [0, M_\ell]$ and $x \mapsto \ell(\theta, (x, y))$ is $\gamma L_c$-Lipschitz for all $\theta \in \Theta$. If Assumptions C and D hold, then with probability at least $1 - e^{-t}$,*

$$\sup_{\theta \in \Theta} |\mathbb{E}_{\widehat{P}_n}[c(T_\gamma(\theta; Z), Z)] - \mathbb{E}_{P_0}[c(T_\gamma(\theta; Z), Z)]| \leq 4D\sqrt{\frac{1}{n}\left(t + \log N\left(\Theta, \frac{[\gamma - L_{xx}]_+ t}{4L_c L_{x\theta}}, \|\cdot\|\right)\right)}. \tag{30}$$

*where $B = L_c M_z$ under assumption (i) and $B = M_\ell/\gamma$ under assumption (ii).*

For both results, the proofs are essentially identical as before, but with an application of Lemma 5 instead of Lemma 1.

# E  PROXIMAL ALGORITHM FOR $\|\cdot\|_\infty$-NORM ROBUSTNESS

In this section, we give a efficient training algorithm that learns to defend against $\|\cdot\|_\infty$-norm perturbations. For simplicity, we assume $\mathcal{Z} = \mathbb{R}^m$ for the rest of this section. Let $\theta \in \Theta$ be some fixed model, $z^0 \in \mathcal{Z}$ a natural example[6] and define $f(z) := \ell(\theta; z)$ to ease notation. Concretely, we are interested in solving the optimization problem

$$\operatorname*{maximize}_z f(z) - \frac{\alpha}{2}\|z - z^0\|_\infty^2$$

Note that this is equivalent to computing the surrogate loss $\phi_\gamma(\theta; z^0) = \sup_{z \in \mathcal{Z}}\{\ell(\theta; z) - \gamma c(z, z^0)\}$ for $c(z, z^0) = \|z - z^0\|_\infty^2$ and $\alpha = 2\gamma$. Our following treatment can easily be modified for the supervised learning scenario $c((x, y), (x^0, y^0)) = \|x - x^0\|_\infty^2 + \infty \cdot \mathbf{1}\{y = y^0\}$ with the convention that $\infty \cdot 0 = 0$. To make our notation consistent with the optimization literature, we consider the minimization problem

$$\operatorname*{minimize}_z -f(z) + \frac{\alpha}{2}\|z - z^0\|_\infty^2. \tag{31}$$

A simple gradient descent algorithm applied to the problem (31) may be slow to converge in practice. Intuitively, this is because the subgradient of $z \mapsto \frac{1}{2}\|z - z^0\|_\infty^2$ is given by $\|z - z^0\|_\infty \cdot s$ where $s$ is a $m$-dimensional vector taking values in $[-1, 1]$ whose coordinates are non-zero only when $|z_j - z_{0,j}| = \|z - z^0\|_\infty$. Hence, at any given iteration of gradient descent, the $\|\cdot\|_\infty$-norm penalty term only gets accounted for by at most a few coordinates.

To remedy this issue, we consider a proximal algorithm for solving the problem (31) (see, for example, Parikh & Boyd (2013) for an comprehensive review of proximal algorithms). For a function $g : \mathcal{Z} \to \mathbb{R}$ and a positive number $\lambda > 0$, the proximal operator for $\lambda g$ is defined by

$$\operatorname{prox}_{\lambda g}(v) := \operatorname*{argmin}_z\left\{g(z) + \frac{1}{2\lambda}\|z - v\|_2^2\right\}.$$

---

[6]We depart from our convention of denoting original datapoints as $z_0$ to ease forthcoming notation.

---

**Algorithm 2** Proximal Algorithm for Maximizing $f(z) - \frac{\alpha}{2}\left\|z - z^0\right\|_\infty^2$

---

INPUT: Stepsizes $\lambda^t$
**for** $t = 0, \dots, T-1$ **do**
    $z^{t+\frac{1}{2}} \leftarrow z^t + \lambda^t \nabla f(z^t)$
    $v^t \leftarrow \text{sort}(|z^{t+\frac{1}{2}} - z^0|, \text{dec})$
    Compute $\beta^t$ as in (33)
    $z^{t+1} \leftarrow z^{t+\frac{1}{2}} - \left[|z^{t+\frac{1}{2}} - z^0| - \beta^t\right]_+ \text{sign}\left(z^{t+\frac{1}{2}} - z^0\right)$

---

Then, the proximal algorithm on the problem (31) consists of two steps at each iteration $t$: $(i)$ for the smooth function $-f(z)$, take a gradient descent step at the current iterate $z^t$ ($z^{t+\frac{1}{2}}$ below) and $(ii)$ for the non-smooth function $\left\|z - z^0\right\|_\infty^2$, take a proximal step for the function $\frac{\lambda^t \alpha}{2}\left\|\cdot - z^0\right\|_\infty^2$ at $z^{t+\frac{1}{2}}$ ($z^{t+1}$ below):

$$z^{t+\frac{1}{2}} = z^t + \lambda^t \nabla f(z^t), \qquad z^{t+1} = \text{prox}_{\frac{\lambda^t \alpha}{2}\|\cdot - z^0\|_\infty^2}\left(z^{t+\frac{1}{2}}\right). \tag{32}$$

The following proposition shows that we can compute the proximal step $z^{t+1}$ efficiently, simply by sorting the vector $|z^{t+\frac{1}{2}} - z^0|$. We denote by $v^t$, the sorted vector of $|z^{t+\frac{1}{2}} - z^0|$ in **decreasing** order. In the proposition, we use the notation $[\cdot]_+ = \max(\cdot, 0)$.

**Proposition 8.** *Define the scale parameter $\beta^t > 0$ by*

$$\beta^t := \frac{1}{1 + \alpha \lambda^t j^t} \sum_{i=1}^{j^t} v_i^t \quad \text{where} \quad j^t := \max\left\{j \in [m] : \sum_{i=1}^{j-1} v_i - \left(\frac{1}{\alpha\lambda^t} + (j-1)\right) v_j < 0\right\}. \tag{33}$$

*Then, $z^{t+1}$ in the proximal update (32) is given by*

$$z^{t+1} = z^{t+\frac{1}{2}} - \left[|z^{t+\frac{1}{2}} - z^0| - \beta^t\right]_+ \text{sign}\left(z^{t+\frac{1}{2}} - z^0\right). \tag{34}$$

See Section E.1 for the proof of the proposition. From the proposition, we obtain the proximal procedure in Algorithm 2 that can be used to solve for the approximate maximizer of $\ell(\theta; z) - \gamma c(z, z^0)$ in Algorithm 1. Heuristically, ignoring the truncation term in the proximal update (34), we have

$$z^{t+1} \approx z^0 + \beta^t \text{sign}(z^t + \lambda^t \nabla f(z^t) - z^0).$$

Here, we move towards the sign of $z^t + \lambda^t \nabla f(z^t) - z^0$ modulated by the term $\beta^t$, as opposed to just the sign of $\nabla f(z^t)$ for the iterated fast sign gradient method (Goodfellow et al., 2015; Kurakin et al., 2016).

### E.1 PROOF OF PROPOSITION 8

In this proof, we drop the subscript on the iteration $t$ to ease notation. We assume without loss of generality that $z^{t+\frac{1}{2}} - z^0 \neq 0$. For some convex, lower semi-continuous function $g : \mathbb{R}^m \to \mathbb{R}$, let $g^*(s) = \sup_s\{s^\top t - g(t)\}$ be the Fenchel conjuagte of $g$. From the Moreau decomposition (Parikh & Boyd, 2013, Section 2.5), we have

$$\text{prox}_g(w) + \text{prox}_{g^*}(w) = w$$

for any $w \in \mathbb{R}^m$. Noting that the conjugate of $z \mapsto \frac{\alpha\lambda}{2}\left\|z - z_0\right\|_\infty^2$ is given by $z \mapsto z^\top z_0 + \frac{1}{2\alpha\lambda}\left\|z\right\|_1^2$, we have

$$\text{prox}_{\frac{\alpha\lambda}{2}\|\cdot - z_0\|_\infty^2}(w) = w - \text{prox}_{\langle z^0, \cdot\rangle + \frac{1}{2\alpha\lambda}\|\cdot\|_1^2}(w) = w - \text{prox}_{\frac{1}{2\alpha\lambda}\|\cdot\|_1^2}(w - z^0)$$

Let us denote the sorted vector (in decreasing order) of $|w - z^0|$ by $v$. Then, in light of the preceeding display, it suffices to show that

$$\text{prox}_{\frac{1}{2\alpha\lambda}\|\cdot\|_1^2}(w - z^0) = [|w - z| - \beta^\star]_+ \text{sign}\left(w - z^0\right) \tag{35}$$

where $\beta^\star$ is defined as in (33). To show that equality (35) holds, note that the first order optimality conditions for

$$\mathrm{prox}_{\frac{1}{2\alpha\lambda}\|\cdot\|_1^2}(w - z^0) = \operatorname*{argmin}_z \left\{ \frac{1}{2}\|z\|_1^2 + \frac{\alpha\lambda}{2}\|z - (w - z^0)\|_2^2 \right\}$$

is given by

$$\|z\|_1 \operatorname{sign}(z_i) + \alpha\lambda(z_i - w_i + z_i^0) = 0 \quad \text{if } |z_i| \neq 0 \tag{36a}$$

$$\|z\|_1 [-1, 1] - \alpha\lambda(w_i - z_i^0) \ni 0 \quad \text{if } |z_i| = 0. \tag{36b}$$

Now, we use the following elementary lemma.

**Lemma 6.** *For $0 \neq v \geq 0$ with decreasing coordinates, the solution to the equation*

$$\sum_{i:v_i>\beta}(v_i - \beta) = \alpha\lambda\beta$$

*exists and is given by*

$$\beta^\star := \frac{1}{1 + \alpha\lambda j^\star}\sum_{i=1}^{j} v_i \quad \text{where } j^\star := \max\left\{ j \in [m] : \sum_{i=1}^{j-1} v_i - \left(\frac{1}{\alpha\lambda} + (j-1)\right)v_j < 0 \right\}.$$

**Proof of Lemma** First, note that $\beta \mapsto \sum_{i:v_i>\beta}(v_i - \beta) - \alpha\lambda\beta =: h(\beta)$ is decreasing. Noting that $\|v\|_1 > 0$ and $-\alpha\lambda\|v\|_\infty < 0$, there exists $\beta'$ such that $h(\beta') = 0$ and $\beta' \in (0, \|v\|_\infty)$. Since $v_i$'s are decreasing and nonnegative, there exists $j'$ such that $v_{j'} > \beta' \geq v_{j'+1}$ (we abuse notation and let $v_{m+1} := 0$). Then, we have

$$\sum_{i=1}^{j'-1}(v_i - v_{j'}) - \alpha\lambda v_{j'} < 0 \leq \sum_{i=1}^{j'}(v_i - v_{j'+1}) - \alpha\lambda v_{j'+1}.$$

That is, $j' = j^\star$. Solving for $\beta'$ in

$$0 = h(\beta') = \sum_{i=1}^{j^\star} v_i - (\alpha\lambda + j^\star)\beta',$$

we obtain $\beta' = \beta^\star$ as claimed. $\qquad\square$

Now, define

$$z^\star = \left[|w - z^0| - \beta^\star\right]_+ \operatorname{sign}\left(w - z^0\right).$$

Then, we have from Lemma 6 that

$$\|z^\star\|_1 = \sum_{i:|w_i - z_i^0|>\beta^\star}(|w_i - z_i^0| - \beta^\star) = \sum_{i=1}^{j^\star}(v_i - \beta^\star) = \alpha\lambda\beta^\star.$$

If $z_i^\star > 0$, then $\operatorname{sign}(z_i^\star) = \operatorname{sign}(w_i - z_i^0)$ so that

$$\|z^\star\|_1 \operatorname{sign}(z_i) + \alpha\lambda(z_i^\star - w_i + z_i^0) = 0.$$

If $z_i^\star = 0$, then $|w_i - z_i^0| \leq \beta^\star$ and

$$\|z^\star\|_1 [-1, 1] - \alpha\lambda(w_i - z_i^0) = \alpha\lambda\beta^\star[-1, 1] - \alpha\lambda(w_i - z_i^0) \ni 0.$$

Hence, $z^\star$ satisfies the optimality condition (36) as desired.

