# OpenReview forum: "Certifying Some Distributional Robustness with Principled Adversarial Training"
_ICLR.cc/2018/Conference — Accept (Oral)_

### Official Review · AnonReviewer2 · 2017-11-27
**a very interesting approach to adversarial training based on robustness over Wasserstein balls**

**Rating:** 9
**Confidence:** 5

**Review:**

This paper proposes a principled methodology to induce distributional robustness in trained neural nets with the purpose of mitigating the impact of adversarial examples. The idea is to train the model to perform well not only with respect to the unknown population distribution, but to perform well on the worst-case distribution in some ball around the population distribution. In particular, the authors adopt the Wasserstein distance to define the ambiguity sets. This allows them to use strong duality results from the literature on distributionally robust optimization and express the empirical minimax problem as a regularized ERM with a different cost. The theoretical results in the paper are supported by experiments.

Overall, this is a very well-written paper that creatively combines a number of interesting ideas to address an important problem.

---

### Official Review · AnonReviewer1 · 2017-11-28
**Adversarial training is an important topic for deep learning, I feel this work may lead to promising principled ways for adversarial training.**

**Rating:** 9
**Confidence:** 4

**Review:**

This paper applies recently developed ideas in the literature of robust optimization, in particular distributionally robust optimization with Wasserstein metric, and showed that under this framework for smooth loss functions when not too much robustness is requested, then the resulting optimization problem is of the same difficulty level as the original one (where the adversarial attack is not concerned). I think the idea is intuitive and reasonable, the result is nice. Although it only holds when light robustness are imposed, but in practice, this seems to be more of the case than say large deviation/adversary exists. As adversarial training is an important topic for deep learning, I feel this work may lead to promising principled ways for adversarial training.

---

### Official Review · AnonReviewer3 · 2017-12-01
**Very interesting principled analysis of robust learning**

**Rating:** 9
**Confidence:** 4

**Review:**

In this very good paper, the objective is to perform robust learning: to minimize not only the risk under some distribution P_0, but also against the worst case distribution in a ball around P_0.

Since the min-max problem is intractable in general, what is actually studied here is a relaxation of the problem: it is possible to give a non-convex dual formulation of the problem. If the duality parameter is large enough, the functions become convex given that the initial losses are smooth.

What follows are certifiable bounds for the risk for robust  learning and stochastic optimization over a ball of distributions. Experiments show that this performs as expected, and gives a good intuition for the reasons why this occurs: separation lines are 'pushed away' from samples, and a margin seems to be increased with this procedure.

---

### Public Comment · (anonymous) · 2017-12-06
**Very cool!**

Very interesting work! I was wondering how the robust MDP/RL setup compares to http://papers.nips.cc/paper/6897-reinforcement-learning-under-model-mismatch.pdf ?

---

> ### Author Response · Authors · 2018-01-05
> **Comparison with Roy et al. (2017)**
>
> Thank you for bringing our attention to Roy et al. (2017). In Section 4.3, we adapted our adversarial training algorithm in the supervised learning setting to reinforcement learning; this approach shares similar motivations as Roy et al. (2017)—and more broadly, the robust MDP literature—where we also solve approximations of the worst-case Bellman equation. Compared to our Wasserstein ball, Roy et al. (2017) uses more simple and tractable worst-case regions. While they give convergence guarantees for their algorithm, the empirical performance of these different worst-case regions remains open.
>
> Another key difference in our experiments is that we assumed access to the simulator for updating the underlying state. This allows us to explore bad regions better. Nevertheless, our adversarial state update in Eqn (20) can be replaced with an adversarial reward update for settings where the simulator cannot be accessed.

---

### Public Comment · ~Aleksander_Madry1 · 2017-12-31
**An interesting attempt but some of the key claims seem to be inaccurate and miss comparison to proper baselines.**

Developing principled approaches to training adversarially robust models is an important (and difficult) challenge. This is especially the case if such an approach is to offer provable guarantees and outperform state of the art methods.

However, after reading this submission, I am confused by some of the key claims and find them to be inaccurate and somewhat exaggerated. In particular, I believe that the following points should be addressed and clarified:

1. The authors claim their methods match or outperform existing methods. However, their evaluations seem to miss some key baselines and parameter regimes.

For example, when reporting the results for l_infty robustness - a canonical evaluation setting in most previous work - the authors plot (in Figure 2b) the robustness only for the perturbations whose size eps (as measured in the l_infty norm) is between 0 and 0.2. (Note that in Figure 2b the x-axis is scaled as roughly 2*eps.) However, in order to properly compare against prior work, one needs to be able to see the scaling for larger perturbations.

In particular, [MMSTV’17] https://arxiv.org/abs/1706.06083 gives a model that exhibits high robustness even for perturbations of l_infty size 0.3. What robustness does the approach proposed in this work offer in that regime?

As I describe below, my main worry is that the theorems in this work only apply for very small perturbations (and, in fact, this seems to be an inherent limitation of the whole approach). Hence, it would be good to see if this is true in practice as well.
In particular, Figure 2b suggests that this method will indeed not work for larger perturbations. I thus wonder in what sense the presented results outperform/match previous work?

After a closer look, it seems that this discrepancy occurs because the authors are reproducing the results of [MMSTV’17] using l_2 based adversarial training.  [MMSTV’17] uses l_infity based training and achieves much better results than those reported in this submission. This artificially handicaps the baseline from [MMSTV’17]. That is, there is a significantly better baseline that is not reflected in Figure 2b. I am not sure why the authors decided to do that.

2. It is hard to properly interpret what actual provable guarantees the proposed techniques offer. More concretely, what is the amount of perturbation that models trained using these techniques are provably robust to?

Based on the presented theorems, it is unclear why they should yield any non-vacuous generalization bounds.

In particular, as far as I can understand, there might be no uniform bound on the amount of perturbation that the trained model will be robust to. This seems to be so as the provided guarantees (see Theorem 4) might give different perturbation resistance for different regions of the underlying distribution. In fact, it could be that for a significant fraction of points we have a (provable) robustness guarantee only for vanishingly small perturbations.

More precisely, note that the proposed approach uses adversarial training that is based on a Lagrangian formulation of finding the worst case perturbation, as opposed to casting this primitive as optimization over an explicitly defined constraint set. These two views are equivalent as long as one has full flexibility in setting the Lagrangian penalization parameter gamma. In particular, for some instances, one needs to set gamma to be *small enough*, i.e., sufficiently small so as it does not exclude norm-eps vectors from the set of considered perturbations. (Here, eps denotes the desired robustness measured in a specific norm such as l_infty, i.e., the prediction of our model should not change under perturbations of magnitude up to eps.)

However, the key point of the proposed approach is to ensure that gamma is always set to be *large enough* so as the optimized function (i.e., the loss + the Lagrangian penalization) becomes concave (and thus provably tractable). Specifically, the authors need gamma to be large enough to counterbalance the (local) smoothness parameter of the loss function.

There seems to be no global (and sufficiently small) bound on this smoothness and, as a result, it is unclear what is the value of the eps-based robustness guarantee offered once gamma is set to be as large as the proposed approach needs it to be.

For the same reason (i.e., the dependence on the smoothness parameter of the loss function that is not explicitly well bounded), the provided generalization bounds - and thus the resulting robustness guarantees - might be vacuous for actual deep learning models.

Is there something I am missing here? If not, what is the exact nature of the provable guarantees that are offered in the proposed work?

---

> ### Author Response · Authors · 2018-01-05
> **[Part II] Our goal is to defend against imperceptible perturbations. More empirical evaluations available.**
>
> 2. Theoretical Guarantees
>
> The motivation for our work is that computing the worst-case perturbation of a deep network under norm-constraints is typically intractable. As we state in the introduction, we simply give up on computing worst-case perturbations at arbitrary budget levels, instead considering small adversarial perturbations. Our theoretical guarantees are concerned with imperceptible changes; we give computational and statistical guarantees for such small (adversarial) perturbations. This is definitely a limit of the approach; given that it is NP hard to certify robustness for larger perturbations this may be challenging to get around.
>
> Our main theoretical guarantee is the certificate of robustness—a data-dependent upper bound on the worst-case performance—given in Theorem 3. This upper bound applies in general, although its efficient computation is only guaranteed for large penalty parameters \gamma and smooth losses. Similarly, as you note, Theorems 2 and 4 only apply in such regimes. To address this, we augment our theoretical guarantees for small adversarial budgets with empirical evaluations in Section 4 and Appendix A. We empirically checked if our level of \gamma = .385 (=.04 * C_2) is above the estimated smoothness parameter at the adversarially trained model and observed that this condition is satisfied on 98% of the training data points.
>
> Our guarantees indeed depend on the problem-dependent smoothness parameter. As with most optimization and statistical learning guarantees, this value is often unknown. This limitation applies to most learning-theoretic results, and we believe that being adaptive to such problem-dependent constants is a meaningful future research direction. With that said, it seems likely (though we have not had time to verify this) that the recent work of Bartlett et al. (https://arxiv.org/pdf/1706.08498.pdf) should apply--it provides covering number bounds our Theorem 3 (Eq. (11-12)) can use.
>
> We hope that our theoretical guarantees are a step towards understanding the performance of these adversarial training procedures. Gaps still remain; we hope future work will close this gap.

---

> ### Author Response · Authors · 2018-01-05
> **[Part I] Our goal is to defend against imperceptible perturbations. More empirical evaluations available.**
>
> Thank you for your interest in our paper. We appreciate the detailed feedback and probing questions.
>
> Upon your suggestions during our meeting at NIPS, we have included a more extensive empirical evaluation of our algorithm. Most notably, we trained and tested our method—alongside other baselines, including [MMSTV17]—on large values of adversarial budgets. We further compared our algorithm trained against L2-norm Lagrangian attacks against other heuristic methods trained against infinity-norm attacks. Lastly, we proposed a (heuristic) proximal variant of our algorithm that learns to defend against infinity-norm attacks. See Appendices A.4, A.5, and E for the relevant exposition and figures.
>
> 1. Empirical Evaluation on Large Adversarial Budgets
>
> Our primary motivation of this paper is to provide a theoretically principled algorithm that can defend against small adversarial perturbations. In particular, we are concerned with provable procedures against small adversarial perturbations that can fool deep nets but are imperceptible to humans. Our main finding in the original empirical experiments in Section 4 was that for such small adversarial perturbations, our principled algorithm matches or outperforms other existing heuristics. (See also point 2 below.)
>
> The adversarial budget epsilon = .3 in the infinity-norm you suggest allows attacks that are highly visible to the human eye. For example, one can construct hand-tuned perturbations that look significantly different from the original image (see https://www.dropbox.com/sh/c6789iwhnooz5po/AABBpU_mg-FRRq7PT1LzI0GAa?dl=0). Defending against such attacks is certainly interesting, but was not our main goal. This probably warrants a longer discussion, but it is not clear to us that infinity-norm-bounded attacks are most appropriate if one allows perceptible image modifications. An L1-budgeted adversary might be able to make small changes in some part of the image, which yields a different set of attacks.
>
> In spite of the departure of large perturbations from our nominal goal of protection against small changes, we test our algorithm on attacks with large adversarial budgets in Appendix A.4. In this case, our algorithm is a heuristic—as are other methods for large adversarial budgets—but we nevertheless match the performance of other methods (FGM, IFGM, PGM) trained against L2-norm adversaries.
>
> Since our computational guarantees are based on strong concavity w.r.t. Lp-norms for p \in (1, 2], our robustly-fitted network defends against L2-norm attacks. Per the suggestion to compare against networks trained to defend against infinity-norm attacks—and we agree, this is an important comparison that we did not perform originally (though we should have)—we compared our method with other heuristics in Appendix A.5.1. On imperceptible L2 and infinity-norm attacks, our algorithm outperforms other heuristics trained to defend against infinity-norm attacks (Figures 11 and 12). On larger attacks, particularly infinity-norm attacks, we observe that other heuristics trained on infinity-norm attacks outperform our method (Figure 12). In this sense, the conclusions we reached from the main figures in our paper—where we considered imperceptible perturbations—are still valid: we match or outperform other heuristic methods for small perturbations.
>
> (Continued in Part II)

---

### Public Comment · (anonymous) · 2018-01-05
**Rigorous approach to Adversarial training but some concerns**

The problems are very well formulated (although only the L2 case is discussed). Identifying a concave surrogate in this mini-max problem is illuminating. The interplay between optimal transport, robust statistics, optimization and learning theory make the work a fairly thorough attempt at this difficult problem. Thanks to the authors for turning many intuitive concepts into rigorous maths. There are some potential concerns, however:

1. The generalization bounds in THM 3, Cor 1, THM 4 for deep neural nets appear to be vacuous, since they scale like \sqrt (d/n), but d > n for deep learning. This is typical, although such generalization bounds are not common in deep adversarial training. So establishing such bounds is still interesting.

2. Deep neural nets generalize well in practice, despite the lack of non-vacuous generalization bounds. Does the proposed WRM adversarial training procedure also generalize despite the vacuous bounds?

In the experimental sections, good performance is achieved at test time. But it would be more convincing if the performance for training data is also shown. The current experiments don't seem to evaluate generalization of the proposed WRM. Furthermore, analysis of other classification problems (cifar10, cifar 100, imagenet) is highly desired.

3. From an algorithmic viewpoint, the change isn't drastic. It appears that it controls the growth of the loss function around the L2 neighbourhood of the data manifold (thanks to the concavity identified). Since L2 geometry has good symmetry, it makes the decision surface more symmetrical between data (Fig 1).

It seems to me that this is the reason for the performance gain at test time, and the size of such \epsilon tube is the robust certificate. So it is unclear how much success is due to the generalization bounds claimed.

I think there is enough contribution in the paper, but I share the opinion of Aleksander Madry, and would like to be corrected for missing some key points.

---

> ### Author Response · Authors · 2018-01-05
> **Addressing concerns**
>
> Thank you for your interest in our paper. We appreciate your detailed feedback.
>
> 1. This is a fair criticism; it seems to apply generally to most learning-theoretic guarantees on deep learning (though see the recent work of Dziugaite and Roy, https://arxiv.org/abs/1703.11008 and Bartlett, Foster, Telgarsky https://arxiv.org/pdf/1706.08498.pdf). We believe that our statistical guarantees in Theorems 3 and 4 are steps towards a principled understanding of adversarial training. Replacing our current covering number arguments with more intricate notions such as margin based-bounds (Bartlett et al. 2017)) would extend the scope of our theoretical guarantees; as Bartlett et al. provide covering number bounds, it seems likely that we could massage them into applying in Theorem 3 (Eqs. (11)-(12)). This is a meaningful future research direction.
>
>
> 2. In Figure 2, we plot our certificate of robustness on two datasets (omitting the statistical error term) and observe that our data-dependent upper bound on the worst-case performance is reasonable. This roughly implies that our adversarial training procedure generalizes, allowing us to learn to defend against attacks on the test set.
>
> “In the experimental sections, good performance is achieved at test time. But it would be more convincing if the performance for training data is also shown. The current experiments don't seem to evaluate generalization of the proposed WRM. Furthermore, analysis of other classification problems (cifar10, cifar 100, imagenet) is highly desired.“
>
> These are both great suggestions. We are currently working on experiments with subsets of Imagenet and will include them in a revision (soon we hope).
>
> 3. Our adversarial training algorithm has intimate connections with other previously proposed heuristics. Our main theoretical contribution is that for small adversarial perturbations, we can show both computational and statistical guarantees for our procedure. More specifically, the computational guarantees for our algorithm are indeed based on the curvature of the L2-norm; provably efficient computation of attacks based on infinity-norms remains open.

---

### Author Response · Authors · 2018-01-05
**Response to Reviews**

We thank the reviewers for their time and positive feedback. We will use the comments and suggestions to improve the quality and presentation the paper. In addition to cleaning up our exposition, we added some content to make our main points more clear. We address these main revisions below.

Our formulation (2) is general enough to include a number of different adversarial training scenarios. In Section 2 (and more thoroughly in Appendix D), we detail how our general theory can be modified in the supervised learning setting so that we learn to defend against adversarial perturbations to only the feature vectors (and not the labels). By suitably modifying the cost function that defines the Wasserstein distance, our formulation further encompasses other variants such as adversarial perturbations only to a fixed small region of an image.

We emphasize that our certificate of robustness given in Theorem 3 applies for any level of robustness \rho. Our results imply that the output of our principled adversarial training procedure has worst-case performance no worse than this data-dependent certificate. Our certificate is efficiently computable, and we plot it in Figure 2 for our experiments. We see that in practice, the bound indeed gives a meaningful performance guarantee against attacks on the unseen test sets.

While the primary focus of our paper is on providing provable defenses against imperceptible adversarial perturbations, we supplement our previous results with a more extensive empirical evaluation. In Appendix A.4, we augment our results by evaluating performance against L2-norm adversarial attacks with larger adversarial budgets (higher values of \rho or \epsilon). Our method also becomes a heuristic for such large values of adversarial budgets, but we nevertheless match the performance of other methods (FGM, IFGM, PGM) trained against L2-norm adversaries. In Appendix A.5.1, we further compare our method——which is trained to defend against L2-norm attacks——with other adversarial training algorithms trained against inf-norm attacks. We also propose a new (heuristic) proximal algorithm for solving our Lagrangian problem with inf-norms, and test its performance against other methods in Appendix A.5.2. In both sections, we observe that our method is competitive with other methods against imperceptible adversarial attacks, and performance starts to degrade as the attacks become visible to the human eye.

Again, we appreciate the reviewers' close reading and thoughtful comments.

---

> ### Public Comment · ~Aleksander_Madry1 · 2018-01-11
> **Re: [Part I] Our goal is to defend against imperceptible perturbations. More empirical evaluations available.**
>
> [I put my reply here as the threads below are now a bit hard to follow.]
>
> Thank you for responding to my comments and making the effort to provide more data. This indeed helps me understand this work better.
>
> I agree that studying the regime of small adversarial perturbation budget epsilon is a very valid research goal. I think, however, that it is important to explicitly mention in the paper that this is the target. Especially, as the proposed methods seem to be inherently restricted to apply to only such a small epsilon regime.
>
> I am not sure though that I agree with the argument why the regime of larger values of epsilon might be less interesting. Yes, some of the larger perturbations will be clearly visible to a human, but some (e.g., the ones that correspond to a change of the background color or its pattern) will not - and we still would like to be robust to them. After all, security guarantees are about getting "for all", not "for some" guarantees.
>
> Now, regarding being explicit about the constants in the bounds, I agree that many optimization and statistical learning guarantees do not provide not provide explicit constants. However, I think the situation in the context considered here is fundamentally different.
>
> After all, for example, in the context of generalization bounds, we always have a meaningful way of checking if a given bound "triggered" for a given model and dataset by testing its performance on a validation/test set. When we talk about robustness guarantee, the whole point is to have it hold even against attacks that we are not able to produce ourselves (but the adversary might). Then, we really need a very concrete guarantee of the form "(With high probability) the model classifies correctly 90% of the test set against perturbation budget of epsilon <= 0.1”.
>
> In the light of this, providing a guarantee of the form "(With high probability) the model correctly classifies 90% of the test set against perturbation budget of some positive epsilon", which is what the proposed guarantees seem to provide, is somewhat less meaningful. (One could argue that, after all, there is always some positive epsilon for which the model is robust.)
>
> It might be worth noting that, e.g., for MNIST, we currently are able to deliver guarantees of the former (explicit) type. For instance, there is a recent work of Kolter and Wong (https://arxiv.org/abs/1711.00851). Although they provide such guarantees via verification techniques and not by proving an explicit generalization bound.
>
> Finally, I am not sure how much bearing the formal NP-hardness of certifying the robustness has here. (I assume you are referring to the result in Appendix B.) Could you elaborate?

---

> > ### Author Response · Authors · 2018-01-24
> > **Response to concerns raised by Madry**
> >
> > Thank you for the detailed follow-up.
> >
> > We will make the point that we deal with imperceptible changes clearer in the paper. We had emphasized that our work is motivated by imperceptible adversarial perturbations from the second paragraph of the paper. We will make this point even clearer and quantify our statements on performance so that there is no confusion that we mainly consider imperceptible changes.
> >
> > As we have noted in our previous response, we agree with you in that robustness to larger perturbations is an important research direction. The point we made in our original response is that infinity-norms may not be the most appropriate norm to consider in this perceptible attack setting. For example, a 1-norm-constrained adversary can change a few pixels in a very meaningful way--with infinity-norms approaching 1--which may be a more suitable model for a perceptible adversary. There are a number of concurrent works on this topic that we believe could lead to more robust learning systems.
> >
> > It is still open whether distributionally robust algorithms (empirically) allow hedging against large adversarial perturbations. At this point, we believe it would be imprudent to call this class of methods “inherently restricted” to the small perturbation regime; indeed, any heuristic method (such as one based on projected gradient descent) has the same restrictions, at least in terms of rigorous guarantees. A more thorough study—on more diverse datasets, model classes and hyperparameter settings—should be conducted in order to draw any meaningful conclusions. We hope to contribute to this effort in the future but we invite others as well, since we believe this is an important question for the community to answer.
> >
> > Our certificate of robustness given in Theorem 3 is efficiently computable for small values of rho, or equivalently, for imperceptible attacks. Hence, this data-dependent certificate provides a upper bound on the worst-case loss so that you are guaranteed to do no worse than this number with high probability. For the achieved level of robustness (rho hat in our notation), our bounds do imply that we are robust to perturbation budgets of this size. Hence, we would argue that Theorem 3 is indeed a flavor of result that satisfies the desiderata you described.
> >
> > There are limitations, and we hope that subsequent work will improve our learning guarantees with a better dependence on model size. This criticism, however, largely applies to most learning-theoretic results applied to deep learning.
> >
> > As we mentioned in our introduction, we agree that recent advances in verification techniques for deep learning are a complementary and important research direction for achieving robust learning systems. Our understanding of these techniques is that they currently have prohibitive computational complexity, even on small datasets such as MNIST. Our results complement these approaches by providing a weaker statistical guarantee with computational effort more comparable to the vanilla training times.
> >
> > The motivation of this paper comes from the fact that formal guarantees on arbitrary levels of robustness is NP-hard. We study the regime of small to moderate levels of robustness to provide guarantees for this regime.

---

### Comment · Area_Chair · 2018-01-24
**Dear authors...**

You have been contacted now by the Area Chair and the Program Chair and asked to respond to comments by the Area Chair. It is imperative that you respond.

---

> ### Author Response · Authors · 2018-01-24
> **Email notification received. Response will be uploaded later today.**
>
> We just received an email notification abut this comment a few minutes ago and somehow did not receive any notification of the original comment uploaded on 21 January. We will upload a response later today.

---

> > ### Comment · Area_Chair · 2018-01-24
> > **OpenReview bug, it seems**
> >
> > Sorry for the rush created by this likely OpenReview bug.  A response today would be most appreciated!

---

> > > ### Author Response · Authors · 2018-01-25
> > > **Response uploaded below.**
> > >
> > > Apologies for the (evidently) tardy response. We have now uploaded a response to the area chair's comments (see below).

---

### Author Response · Authors · 2018-02-25
**Updated paper**

Thanks to the reviewers, AC, and senior PC for the valuable feedback; the controversy around the paper has certainly been elucidating. To reflect our understanding of the results in the paper, which we hope others share, we have updated our paper’s title and introduction to make more explicit the claims (and concomitant limitations) of the work, better situating it in relation to other research. It seems that there was confusion about the results in the paper, and we have tried to clear up that our approach seeks a sweet spot balancing the goals of computational efficiency and certification of robustness. In the original submission, we noted in the introduction that "The key feature of the penalty problem (2) is that moderate levels of robustness are achievable at essentially no computational or statistical cost for smooth losses $\ell$.” We’ve tried to make this facet of the work more apparent throughout the paper, noting the connection between small perturbations/imperceptible changes to moderate levels of robustness. We believe this paper is a step towards efficiently building certifiably stable deep networks, though of course substantial work remains. We note a few possible directions for future research in the paper along with explicit limitations of our work. It is plausible to us that principled guarantees, even for complex deep networks, are possible with reasonable computational tools, and we hope others will continue to focus on these efforts.

---

### Author Response · Authors · 2018-12-19
**Updated paper**

We have fixed some typos in the main text as well as the appendices. Thanks to members of the community for their feedback and interest in our paper.

---

### Public Comment · (anonymous) · 2019-05-31
**Questions regarding \Theta**

Hi, I found your paper to be very interesting and have some questions regarding the implementation.

1. How did you determine the constrained set \Theta and what does it mean in terms of implementation? Is it just weight clipping? If yes, what range do you use to clip?

2. When solving the inner maximization in Algorithm 1, how did you use \epsilon?

3. What does WRM stand for?

Thank you!

---

### Decision · Program_Chairs · 2018-01-29
**ICLR 2018 Conference Acceptance Decision**

**Decision:**

Accept (Oral)

**Comment:**

This paper attracted strong praise from the reviewers, who felt that it was of high quality and originality.  The broad problem that is being tackled is clearly of great importance.

This paper also attracted the attention of outside experts, who were more skeptical of the claims made by the paper. The technical merits do not seem to be in question, but rather, their interpretation/application. The perception by a community as to whether an important problem has been essentially solved can affect the choices made by other reviewers when they decide what work to pursue themselves, evaluate grants, etc. It's important that claims be conservative and highlight the ways in which the present work does not fully address the broader problem of adversarial examples.

Ultimately, it has been decided that the paper will be of great interest to the community. The authors have also been entrusted with the responsibility to consider the issues raised by the outside expert (and then echoed by the AC) in their final revisions.

One final note: In their responses to the outside expert, the authors several times remark that the guarantees made in the paper are, in form, no different from standard learning-theoretic claims: "This criticism, however, applies to many learning-theoretic results (including those applied in deep learning)." I don't find any comfort in this statement. Learning theorists have often focused on the form of the bounds (sqrt(m) dependence and, say, independence from the # of weights) and then they resort to empirical observations of correlation to demonstrate that the value of the bound is predictive for generalization. because the bounds are often meaningless ("vacuous") when evaluated on real data sets. (There are some recent examples bucking this trend.) In a sense, learning theorists have gotten off easy. Adversarial examples, however, concern security, and so there is more at stake. The slack we might afford learning theorists is not appropriate in this new context. I would encourage the authors to clearly explain any remaining work that needs to be done to move from "good enough for learning theory" to "good enough for security". The authors promise to outline important future work / open problems for the community. I definitely encourage this.